# Rethinking Sparse Mixture of Experts from a Unified Perspective

**Giang Do** [1] **Hung Le** [1] **Truyen Tran** [1]

## Abstract

Sparse Mixture of Experts (SMoE) models scale the capacity of models while maintaining constant computational overhead. SMoE methods fall into two categories: *Token Choice*, which routes each token to a fixed number of experts, and *Expert Choice*, which assigns a fixed number of tokens to each expert. However, the use of fixed budgets for tokens or experts causes both approaches to select irrelevant token-expert pairs or overlook critical assignments, which degrades overall performance. To fill that gap, we rethink SMoE from a *unified perspective* through the lens of *linear programming*, which provides a general formulation for SMoE models. Furthermore, we introduce **Unified Sparse Mixture of Experts (USMoE)**, a novel framework comprising a *unified mechanism* and a *unified score* to overcome these limitations. We provide both theoretical justification and empirical evidence demonstrating USMoE's effectiveness. Extensive evaluations across diverse data settings (clean and corrupted), multiple domains (including texts and vision tasks), and different learning approaches (training-free and training-based) show that USMoE not only delivers significant performance improvements over existing SMoE methods, but also enables more flexible expert selection budgets, reducing inference costs without compromising model performance. Our implementation is publicly available at https://github.com/giangdip2410/USMoE.

## 1. Introduction

Sparse Mixture of Experts (SMoE) models have achieved notable success in natural language processing (NLP) and

[1]Applied Artificial Intelligence Intiative (A2I2), Deakin University, Victoria, Australia. Correspondence to: Giang Do <truong.do@deakin.edu.au>.

*Proceedings of the 43$^{rd}$ International Conference on Machine Learning*, Seoul, South Korea. PMLR 306, 2026. Copyright 2026 by the author(s).

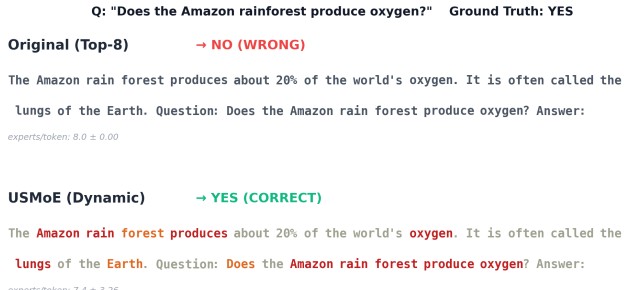

Q: "Does the Amazon rainforest produce oxygen?"   Ground Truth: YES

**Original (Top-8)**          → NO (WRONG)

The Amazon rain forest produces about 20% of the world's oxygen. It is often called the lungs of the Earth. Question: Does the Amazon rain forest produce oxygen? Answer:
*experts/token: 8.0 ± 0.00*

**USMoE (Dynamic)**          → YES (CORRECT)

The Amazon rain forest produces about 20% of the world's oxygen. It is often called the lungs of the Earth. Question: Does the Amazon rain forest produce oxygen? Answer:
*experts/token: 7.4 ± 3.26*

*Figure 1.* Illustration of USMoE compared with Qwen3-30B-A3B-Instruct (Original) on the BoolQ dataset. We highlight the key difference between the two approaches: the Original model adopts a Token Choice strategy with a fixed number of experts per token, whereas USMoE dynamically selects the number of experts per token, allowing the model to focus on more relevant tokens and thereby improving performance. Best viewed in color.

visual representation learning tasks (Du et al., 2022; Fedus et al., 2022; Riquelme et al., 2021; Shen et al., 2023). These advancements build on the Transformer architecture (Vaswani et al., 2017) and its variants (Child et al., 2019; Dai et al., 2019), which leverage large datasets and significant compute resources. However, training large Transformer models can be prohibitively expensive, requiring extensive compute hours (Kaddour et al., 2023). To overcome this issue, SMoE models activate only a subset of experts for each input, reducing inference time compared to dense models (Shazeer et al., 2017; Zoph et al., 2022; Artetxe et al., 2022; Ludziejewski et al., 2024). The SMoE architecture can be categorized into two variants: *Token Choice*, which assigns experts to each token (Dai et al., 2024; Team, 2024; Muennighoff et al., 2025; Jiang et al., 2024a), and *Expert Choice*, which assigns tokens to each expert (Zhou et al., 2022). The advantage of Token Choice lies in its ability to select experts for each token, while Expert Choice ensures a more balanced token distribution across experts.

Despite their promising results, SMoE models have several limitations. The Expert Choice approach suffers from token dropping (Zhou et al., 2022), while the Token Choice approach struggles with unbalanced expert loading (Shazeer et al., 2017). The use of fixed budgets for tokens or experts causes both approaches to select irrelevant token–expert pairs or overlook critical assignments. Additionally, projecting high-dimensional representations into a

lower-dimensional routing space is prone to representation collapse, where routing decisions become skewed toward a small subset of experts or multiple experts converge to similar representations (Chi et al., 2022; Chen et al., 2022). Recent research has explored improving router policies (Chi et al., 2022; Chen et al., 2023; Do et al., 2023) to mitigate these issues. However, existing methods face three key challenges: (1) The reliance on auxiliary losses requires careful balancing between the router loss and the task loss, which introduces trade-offs; (2) *Token Choice* (TC) struggles to handle noisy tokens effectively; and (3) *Expert Choice* (EC) suffers from information leakage issues, which significantly degrade performance on autoregressive models (Zhou et al., 2022; Wang et al., 2024; Raposo et al., 2024). As a result, the question of how to optimally select experts or tokens remains open.

In this paper, we revisit SMoEs through *a unified perspective* to better understand the criteria for expert and token selection. From this perspective, Token Choice selects experts along the expert dimension by assigning each token to its most similar expert, while Expert Choice selects tokens along the token dimension by allowing each expert to choose the most similar tokens. This view highlights a key trade-off: Expert Choice risks dropping important tokens, whereas Token Choice struggles to handle noisy or irrelevant tokens. Furthermore, both approaches suffer from the problem of representation collapse (Chi et al., 2022; Do et al., 2023; Pham et al., 2024).

Building on this analysis, we propose the Unified Sparse Mixture of Experts (USMoE), a robust and efficient framework consisting of two key components: (1) the Unified Score and (2) the Unified Mechanism. The Unified Score is defined as a linear combination of two mapping functions that transform the dot product similarity between token and expert embeddings into a probability distribution. In parallel, the Unified Mechanism incorporates information from both the token and expert dimensions, as illustrated in Figure 2. These components enable the SMoE model to dynamically prioritize tokens or experts while ensuring the selection of the most similar token-expert pair, enhancing both robustness and effectiveness. We provide both theoretical justification and empirical evidence demonstrating USMoE's effectiveness. Extensive evaluations across diverse data settings (clean and corrupted), multiple domains (including texts and vision tasks), and different learning approaches (training-free and training-based) show that US-MoE not only delivers significant performance improvements over existing SMoE methods, but also enables more flexible expert selection budgets, reducing inference costs without compromising model performance.

**Key contributions:** (1) We present a *unified perspective* on Sparse Mixture of Experts (SMoE) that fomulates con-

ventional routers as a *linear programming* problem, and propose *USMoE*, a robust and efficient framework built upon a *Unified Score* and a *Unified Mechanism*, offering improved efficiency, robustness, and flexibility over existing approaches. (2) We **theoretically show** that USMoE yields more stable and robust expert selection compared to baseline routing methods. (3) We conduct **extensive experiments** across large language and vision models, spanning both training-free and training-based settings, to comprehensively evaluate the effectiveness, robustness, and flexibility of USMoE. (4) We provide **in-depth analyses** that elucidate why USMoE works and demonstrate its improved interpretability and reliability.

## 2. Related Work

**Sparse Mixture of Experts (SMoE).** Sparse Mixture of Experts (SMoE), an extension of the Mixture of Experts framework (Jacobs et al., 1991; Jordan & Jacobs, 1994), has gained traction with large language models and has since been applied in various domains, including computer vision and speech recognition (Zhou et al., 2022; Riquelme et al., 2021). The SMoE architecture consists of two main variants: **Token Choice**, where experts are assigned to each token (Shazeer et al., 2017; Fedus et al., 2022; Jiang et al., 2024a; Do et al., 2025b), and **Expert Choice**, where tokens are assigned to specific experts (Zhou et al., 2022). Further discussion of related work can be found in Section A.2.

## 3. Methodology

We introduce Unified Sparse Mixture of Experts (USMoE), a novel and efficient Sparse Mixture of Experts framework designed to address the limitations of both Token Choice and Expert Choice through a Unified Score, a scoring function that balances expert and token important and Unified Mechanism, a structured routing strategy that consider information from both token dimension and expert dimension.

### 3.1. Preliminaries

The Sparse Mixture of Experts (SMoE) architecture replaces the MLP layers in standard transformers with Mixture of Experts (MoE) layers (Shazeer et al., 2017). Let $h \in \mathbb{R}^{T \times d}$ denote the token representations from the attention layer, where $T$ is the number of tokens and $d$ is the hidden dimension. Given $N$ expert functions $\{\text{FFN}_j\}_{j=1}^{N}$, a gating projection matrix $W \in \mathbb{R}^{d \times N}$ maps tokens to expert affinity scores. Denotes $S = f(hW) \in \mathbb{R}^{T \times N}$ is a compatibility score, where $f$ maps the scores to a routing distribution (e.g., softmax or sigmoid). Given $X \in \{0,1\}^{T \times N}$ be a binary routing matrix, where $x_{ij} = 1$ if token $i$ is routed to expert $j$, the output of the SMoE layer is:

$$f_{\text{SMoE}}(h) = \sum_{j=1}^{N} S_{\cdot j} \odot X_{\cdot j} \odot \text{FFN}_j(h \odot X_{\cdot j}), \quad (1)$$

where $\text{FFN}_j(\cdot)$ denotes the computation performed by expert $j^{th}$, and $\odot$ represents element-wise multiplication, and FFN denotes for the Feed-forward neural networks.

### 3.2. Experts Selection as a Unified Perspective

Viewing expert selection as a linear programming problem allows for globally optimizing expert assignments by maximizing similarity scores under a strict sparsity (computational) constraint.

**Expert Selection Optimization.** Given a computational budget of $c$ experts per token, the objective of Sparse Mixture-of-Experts (MoE) routing is to determine the optimal expert assignment $X^*$ that maximizes the total compatibility score, subject to a sparsity constraint $c$:

$$
\begin{aligned}
\text{maximize} \quad & \sum_{i=1}^{T} \sum_{j=1}^{N} S_{ij} x_{ij} \\
\text{subject to} \quad & \sum_{i=1}^{T} \sum_{j=1}^{N} x_{ij} \leq c \\
& x_{ij} \in \{0, 1\}, \quad \forall i, j.
\end{aligned}
\quad (2)
$$

**Token-Choice (TC).** TC assumes uniform importance across tokens, assigning each token $i$ to a fixed number of experts $k = \lfloor c/T \rfloor$. This adds the per-token constraint:

$$\sum_{j=1}^{N} x_{ij} = k, \quad \forall i \in 1, \dots, T. \quad (3)$$

**Expert-Choice (EC).** Conversely, EC ensures uniform expert utilization by allowing each expert $j$ to select the top $e = \lfloor c/N \rfloor$ tokens. This is modeled by constraining the per-expert capacity:

$$\sum_{i=1}^{T} x_{ij} = e, \quad \forall j \in 1, \dots, N. \quad (4)$$

**Load-Balancing (LB).** LB relaxes the strict equality of EC, seeking to distribute tokens approximately evenly to prevent expert collapse. It introduces a soft constraint or auxiliary loss such that:

$$\sum_{i=1}^{T} x_{ij} \approx \lfloor c/N \rfloor, \quad \forall j \in 1, \dots, N. \quad (5)$$

**Solution.** To solve the optimization problem in Equation. 2, we observe that selecting $c$ binary variables $x_{ij}$ to maximize the total score is equivalent to choosing the $c$ largest entries of the score matrix $S \in \mathbb{R}^{T \times N}$.

Let $\text{vec}(S) \in \mathbb{R}^{TN}$ denote the vectorization of $S$, obtained by flattening the matrix in row-major order. Define the operator $\text{argtopk}(\mathbf{v}, c)$ as the set of indices of the $c$ largest elements in vector $\mathbf{v}$:

$$\Omega^* = \text{argtopk}\big(\text{vec}(S), c\big). \quad (6)$$

Let $\text{idx}(i, j)$ denote the mapping from 2D coordinates $(i, j)$ to the corresponding 1D index in $\text{vec}(S)$. The optimal binary assignment $X^* \in \{0, 1\}^{T \times N}$ is then:

$$
x_{ij}^* = \begin{cases} 1, & \text{if } \text{idx}(i, j) \in \Omega^*, \\ 0, & \text{otherwise.} \end{cases}
\quad (7)
$$

**Definition 3.1** (Unified Score Function). Let $S \in \mathbb{R}^{T \times N}$ be the compatibility score matrix between $T$ tokens and $N$ experts. Define two score mapping functions:

- $S_t \in \mathbb{R}^{T \times N}$: a row-wise scoring function used in Token Choice (e.g., softmax applied across each row).

- $S_e \in \mathbb{R}^{T \times N}$: a column-wise scoring function used in Expert Choice (e.g., softmax applied across each column).

The *Unified Score Function* combines both perspectives via a linear combination:

$$S_{\text{USMoE}} = \alpha \cdot S_e + \beta \cdot S_t,$$

where $\alpha, \beta \in \mathbb{R}$ are non-negative coefficients such that $\alpha + \beta = 1$.

*Remark* 3.2. The Unified Score as Definition 3.1 integrates both token-centric and expert-centric preferences to inform more balanced routing decisions.

**Proposition 3.3.** *Let $S \in \mathbb{R}^{T \times N}$ be the compatibility score matrix and $c \in \mathbb{N}$ a global routing budget. Consider the objective:*

$$M = \langle S, X \rangle = \sum_{i=1}^{T} \sum_{j=1}^{N} S_{ij} x_{ij},$$

*subject to the constraint $\sum_{i,j} x_{ij} \leq c$, with $x_{ij} \in \{0, 1\}$.*

*Let $X_{USMoE} = TopK(S, c)$ be the binary mask produced by selecting the top-$c$ entries of $S$ globally. Then for any other feasible binary routing matrix $X_T \in \{0, 1\}^{T \times N}$ satisfying the same budget constraint $\sum_{i,j} (X_T)_{ij} \leq c$, we have:*

$$\langle S, X_T \rangle \leq \langle S, X_{USMoE} \rangle.$$

*Remark* 3.4. The Proposition 3.3 indicate that the Unified Mechanism (USMoE) yields the optimal solution that maximizes the total similarity under the global budget constraint.

**Lemma 3.5** (Robustness to Noisy Tokens). *Let $S \in \mathbb{R}^{T \times N}$ be the compatibility score matrix between $T$ tokens and $N$ experts, and suppose a subset of tokens $\mathcal{N} \subset \{1, \ldots, T\}$ are corrupted such that their score vectors contain additive noise: $S_{i,:} = S^*_{i,:} + \epsilon_i$ for $i \in \mathcal{N}$, where $S^*$ denotes the clean score matrix and $\|\epsilon_i\|_\infty \leq \delta$.*

*Let $j^*_i = \arg\max_j S^*_{ij}$ denote the correct expert for token $i$ under clean scores, and define the score margin:*

$$\Delta^{(i)}_{\min} = \min_{j \neq j^*_i} \left( S^*_{ij^*_i} - S^*_{ij} \right).$$

*Then:*

1. *Under Token Choice, a noisy token $i \in \mathcal{N}$ is misrouted whenever $\Delta^{(i)}_{\min} < 2\delta$, depending solely on the row-wise noise perturbation.*

2. *Under the Unified Mechanism (USMoE) with global top-$c$ selection, a noisy token-expert pair $(i, j)$ must rank highly among **all** $T \times N$ entries. This global competition provides an additional filtering effect: even if noise elevates $S_{ij}$ within row $i$, the pair $(i, j)$ is only selected if it also surpasses entries from other (clean) tokens.*

*Consequently, the misrouting probability satisfies:*

$$P^{USMoE}_{mis} \leq P^{TC}_{mis},$$

*with strict inequality when clean tokens provide sufficient competition in the global ranking.*

*Remark* 3.6. The robustness advantage of USMoE arises from its global selection mechanism. Unlike Token Choice, which performs routing independently for each token, US-MoE enforces competition among all token-expert pairs across the entire score matrix. This global competition serves as a natural filtering effect, whereby clean token-expert pairs with high affinity scores effectively crowd out noise amplified entries from the top-$c$ selection.

**Lemma 3.7.** *The Unified Score Function $S_{USMoE} = \alpha \cdot S_e + \beta \cdot S_t$, where $\alpha, \beta \in \mathbb{R}$ are non-negative coefficients such that $\alpha + \beta = 1$, is more robust to representation collapse (Chi et al., 2022) than using $S_t$ or $S_e$ alone.*

*Remark* 3.8. Lemma 3.7 establishes that USMoE outperforms conventional SMoEs by mitigating the representation collapse issue, as discussed in (Chi et al., 2022).

## 3.3. Unified Sparse Mixture of Experts (USMoE)

**Unified Score.** We empirically observe that *Token Choice* effectively captures in-context learning behavior, while *Expert Choice* conveys rich semantic alignment as Figure 8. To harness the complementary strengths of both, we propose the *Unified Score*, defined in Algorithm 1, as a weighted sum of the Token Choice and Expert Choice scores. This unified formulation integrates both token-centric and expert-centric signals to enable more balanced and informed routing decisions. Empirically, we find that setting the combination weight to $\alpha \approx 0.5$ yields a robust trade-off between the two strategies.

**Unified Mechanism.** We propose the *Unified Mechanism*, which formulates expert-token assignment as a joint selection problem. By flattening the similarity matrix and selecting the top-$N$ expert-token pairs based on the highest unified scores, as illustrated in Figure 2, this mechanism enables efficient, context-aware routing within sparse MoE architectures.

## 4. Experiments

In this section, we evaluate our method across both Large Language Models (LLMs) and Vision tasks, under clean and adversarial (attack) settings, and in both training-free and training scenarios. We empirically demonstrate the advantages of USMoE over Token Choice (TC) and Expert Choice (EC) across advanced Sparse Mixture of Experts (SMoE) models, including QwenMoE (Team, 2024), OL-MoE (Muennighoff et al., 2025), and DeepSeekMoE (Dai et al., 2024). Through extensive experiments, we show that: (1) USMoE outperforms baseline methods even without additional computational cost; (2) USMoE provides significant improvements across both language and vision domains; (3) Our method demonstrates robustness in both LLMs and vision tasks; and (4) USMoE supports flexible Top-k expert selection, which is valuable for scenarios with limited computational resources while maintaining competitive performance.

### 4.1. LLMs Training-free Evaluation

#### 4.1.1. LARGE REASONING MODELS

**Settings.** We evaluate our approach on two state-of-the-art models from the Qwen3-MoE family (Team, 2025): *Qwen3-30B-A3B-Thinking* and *Qwen3-30B-A3B-Instruct*. Both models comprise 48 layers with 128 experts, of which 8 are activated per token. As baselines, we consider two Token Choice methods, the original models and MoEE (Li & Zhou, 2025), as well as an Expert Choice routing method (Zhou et al., 2022). All methods are evaluated in a zero-shot setting under both clean conditions and corrupted environments, where we inject *15%* random Gaussian noise to assess ro-

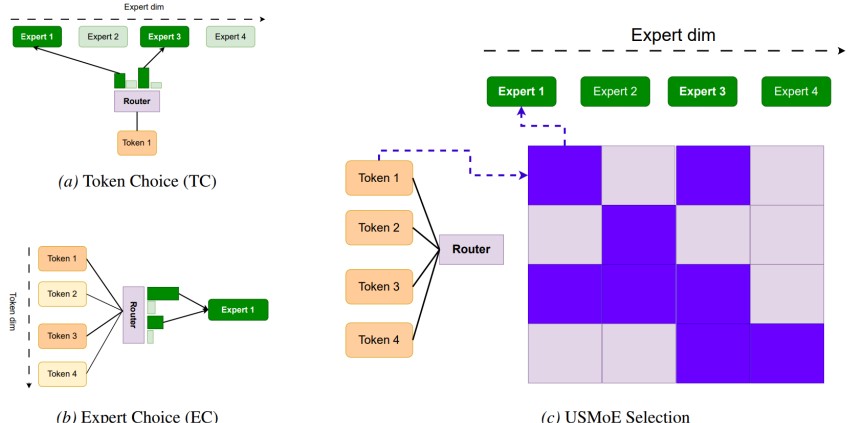

*Figure 2.* Illustration of the USMoE, which jointly considers both token and expert dimensions. *Token Choice* (TC; a) adopts a token-centric strategy that operates only along the expert dimension, while *Expert Choice* (EC; b) follows an expert-centric strategy that focuses solely on the token dimension. Both approaches struggle to effectively exclude irrelevant token-expert pairs. In contrast, USMoE employs a *unified perspective* by formulating selection as a two-dimensional process over both token and expert dimensions, enabling the identification of globally optimal token-expert matches while suppressing irrelevant pairs.

bustness. We further benchmark USMoE on six widely used reasoning datasets: ARC-Challenge and ARC-Easy (Clark et al., 2018) for scientific question answering at different difficulty levels; BoolQ (Clark et al., 2019) for binary reading comprehension; OpenBookQA (Mihaylov et al., 2018) for reasoning over supporting facts; PIQA (Bisk et al., 2020) for physical commonsense reasoning; and WinoGrande (Sakaguchi et al., 2021) for challenging coreference resolution.

**Key Results.** As shown in Table 1, USMoE consistently outperforms the original approach across all six datasets in the zero-shot setting under both clean and corrupt conditions. Specifically, for **Qwen3-30B-A3B-Thinking**, US-MoE achieves an average improvement of **1.5%** over the original model, outperforming all baselines on every dataset. For **Qwen3-30B-A3B-Instruct**, USMoE yields an average gain of **0.8%** in the clean setting. More notably, under corrupt conditions, USMoE substantially enhances robustness, surpassing the original models by an average of **12.4%** for the Thinking variant and **13.5%** for the Instruct variant without any additional training cost. These results demonstrate that USMoE not only improves standard performance but also significantly strengthens model robustness against noisy inputs, such as user typos, at inference time.

**USMoE Reduces Router Sensitivity.** To evaluate the robustness of USMoE relative to baseline methods, we measure the impact of noise by comparing expert selection decisions under clean and corrupt settings. We report the resulting router sensitivity scores in Figure 3. Despite being subjected to the same 15% noise attack as the baselines, USMoE consistently achieves the **lowest sensitivity**, with scores approximately **half** those of competing methods. These results indicate that USMoE yields a substantially more noise-robust routing representation, making expert selection markedly less sensitive to input perturbations. The empirical results are **consistent with Lemma 3.5**, which we propose and prove in the theoretical analysis.

**USMoE Enables More Flexible TopK Selection.** Beyond

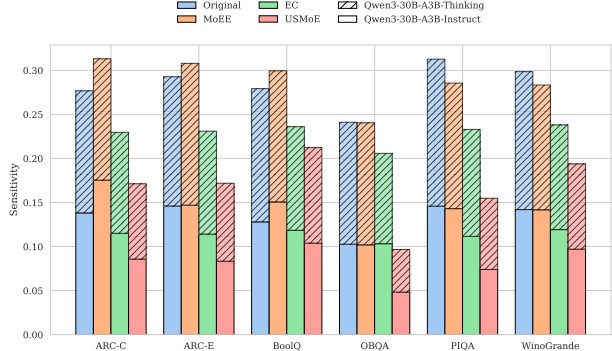

*Figure 3.* Sensitivity comparison of routing methods across multiple reasoning benchmarks. USMoE consistently exhibits lower sensitivity than Original, MoEE, and EC for both Instruct and Thinking variants of Qwen3-30B-A3B. Lower is better. Best viewed in color.

improved routing robustness, USMoE also supports a more flexible Top-$k$ selection mechanism. By converting the TopK parameter from Token Choice into an equivalent form $\lfloor k \cdot L \rfloor$, where $k$ denotes the Token Choice TopK ratio and $L$ is the sequence length, USMoE naturally accommodates both fractional and integer Top-$k$ settings. This flexibility allows USMoE to generalize across different routing configurations without modification. As shown in Figure 4, USMoE achieves performance parity with the original Qwen3-Instruct model on BoolQ using only $k = 3.5$ compared to the baseline $k = 8$. These findings, obtained under corrupted settings, highlight USMoE's ability to sustain high accuracy with substantially lower computational costs through the use of fractional Top-K selection.

### 4.1.2. TEXT EMBEDDING EVALUATION

**Settings.** In this section, inspired by (Li & Zhou, 2025), we test our method as a plug-in framework on well-trained SMoE models, including **OLMoE-1B-7B** (Muennighoff et al., 2025):, **DeepSeekMoE-16B** (Dai et al., 2024),

| Benchmark | Clean Setting | | | | Corrupt Setting | | | |
| | TC | | EC | USMoE | TC | | EC | USMoE |
| | Original | MoEE | | | Original | MoEE | | |
|---|---|---|---|---|---|---|---|---|
| *Qwen3-30B-A3B-Thinking* | | | | | | | | |
| ARC-C | 0.584 ± .014 | 0.275 ± .013 | 0.470 ± .015 | **0.600** ± .014 | 0.244 ± .013 | 0.279 ± .013 | 0.353 ± .014 | **0.383** ± .014 |
| ARC-E | 0.819 ± .008 | 0.241 ± .009 | 0.670 ± .010 | **0.822** ± .008 | 0.341 ± .010 | 0.237 ± .009 | 0.442 ± .010 | **0.558** ± .010 |
| BoolQ | 0.866 ± .006 | 0.378 ± .008 | 0.785 ± .007 | **0.870** ± .006 | 0.556 ± .009 | 0.378 ± .008 | 0.627 ± .008 | **0.727** ± .008 |
| OBQA | 0.424 ± .022 | 0.292 ± .020 | 0.378 ± .022 | **0.450** ± .022 | 0.294 ± .020 | 0.300 ± .021 | 0.322 ± .021 | **0.344** ± .020 |
| PIQA | 0.812 ± .009 | 0.517 ± .012 | 0.649 ± .011 | **0.842** ± .009 | 0.520 ± .012 | 0.530 ± .012 | 0.554 ± .012 | **0.619** ± .011 |
| WinoGrande | 0.729 ± .012 | 0.505 ± .014 | 0.596 ± .014 | **0.739** ± .012 | 0.490 ± .014 | 0.522 ± .014 | 0.513 ± .014 | **0.560** ± .014 |
| Average | 0.706 ± .012 | 0.368 ± .013 | 0.591 ± .013 | **0.721** ± .012 | 0.408 ± .013 | 0.374 ± .013 | 0.469 ± .013 | **0.532** ± .013 |
| *Qwen3-30B-A3B-Instruct* | | | | | | | | |
| ARC-C | 0.631 ± .014 | 0.270 ± .013 | 0.512 ± .015 | **0.646** ± .014 | 0.253 ± .013 | 0.275 ± .013 | 0.354 ± .014 | **0.433** ± .014 |
| ARC-E | 0.838 ± .008 | 0.239 ± .009 | 0.702 ± .009 | **0.846** ± .007 | 0.361 ± .010 | 0.239 ± .009 | 0.483 ± .010 | **0.609** ± .010 |
| BoolQ | 0.886 ± .006 | 0.378 ± .008 | 0.853 ± .006 | **0.894** ± .005 | 0.600 ± .009 | 0.378 ± .008 | 0.747 ± .008 | **0.808** ± .007 |
| OBQA | 0.454 ± .022 | 0.292 ± .020 | 0.374 ± .022 | **0.460** ± .022 | 0.300 ± .021 | 0.302 ± .021 | 0.326 ± .021 | **0.336** ± .021 |
| PIQA | 0.805 ± .009 | 0.521 ± .012 | 0.646 ± .011 | **0.810** ± .009 | 0.516 ± .012 | 0.530 ± .012 | 0.568 ± .012 | **0.605** ± .011 |
| WinoGrande | 0.733 ± .012 | 0.497 ± .014 | 0.583 ± .014 | **0.736** ± .012 | 0.494 ± .014 | 0.518 ± .014 | 0.513 ± .014 | **0.545** ± .014 |
| Average | 0.724 ± .012 | 0.366 ± .013 | 0.612 ± .013 | **0.732** ± .012 | 0.421 ± .013 | 0.374 ± .013 | 0.499 ± .013 | **0.556** ± .013 |

*Table 1.* Performance comparison of various methods for Qwen3-30B-A3B-Thinking and Qwen3-30B-A3B-Instruct using 0-shot evaluation across six reasoning benchmarks. We evaluate both clean and corrupt settings, grouping methods by selection strategy: **TC** (Token Choice), **EC** (Expert Choice), and our proposed **USMoE**. The best results are highlighted in **bold**.

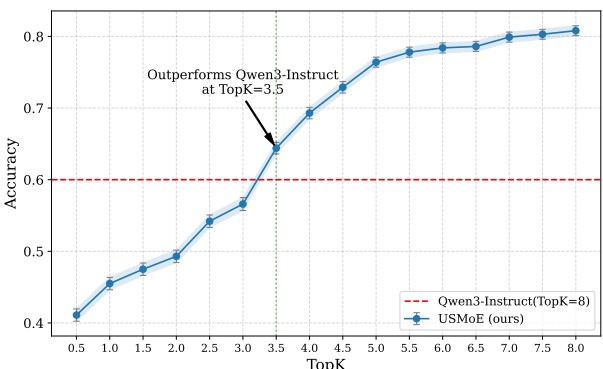

*Figure 4.* Demonstration of the flexibility of USMoE in supporting both fractional and non-fractional Top-$K$ routing, compared with Token Choice, on Qwen3-30B-A3B-Instruct (abbreviated as Qwen3-Instruct) evaluated on the BoolQ dataset under corrupted settings. USMoE outperforms the original Qwen3-30B-A3B-Instruct starting from Top-$K = 3.5$, enabling reduced computational cost while maintaining competitive performance. Best viewed in color.

**Qwen1.5-MoE-A2.7B** (Team, 2024). We evaluate performance on a subset of tasks from the Massive Text Embedding Benchmark (MTEB) (Muennighoff et al., 2023), which covers key downstream applications for sentence embeddings, including Classification, Clustering, Pair Classification, Re-ranking, Retrieval, Semantic Textual Similarity (STS), and Summarization. Following the MTEB evaluation framework, we use Accuracy for Classification, V-Measure for Clustering, Average Precision for Pair Classification, Mean Average Precision for Re-ranking, nDCG for Retrieval, and Spearman's correlation for STS and Summarization.

**Key Results.** USMoE significantly outperforms standard routing baselines across a range of MTEB tasks with PromptEOL (Jiang et al., 2024b), as shown in Table 2. Compared to the Token Choice (TC) approach, our method achieves average performance improvements of 10.8%, 17.6%, and 18.2% for the OLMoE, Qwen1.5-MoE, and DeepSeekMoE architectures, respectively. Notably, USMoE maintains a clear advantage over both TC and Expert Choice (EC) frameworks, with particularly strong gains in Semantic Textual Similarity (STS) and Reranking. These results suggest that USMoE provides a more effective and stable expert activation strategy without the need for further fine-tuning. For a full breakdown of task-specific performance, refer to Appendix A.

### 4.2. Training from Scratch

#### 4.2.1. LANGUAGE MODELS

**Settings.** To assess the effectiveness of our method, we compare USMoE with the Token Choice approaches, including SMoE (Jiang et al., 2024a), SMoE-Dropout (abbreviated as "SMoE-DR"), XMoE (Chi et al., 2022), and StableMoE (Dai et al., 2022), as well as the Expert Choice approach (Zhou et al., 2022) for pre-training and fine-tuning tasks. We follow the approach of (Chen et al., 2023) and use a base Transformer-XL (Dai et al., 2019) with four decoder layers. We train both base and large-scale versions of Transformer-XL on four datasets (Enwik8, Text8, Wikitext-

| Model | Task | Router | TC | EC | MoEE | USMoE |
|---|---|---|---|---|---|---|
| | Classification | 43.1 | 57.7 | 56.2 | 51.7 | **61.4** |
| | Clustering | 16.2 | 24.8 | 26.9 | 23.2 | **32.1** |
| | PairClassification | 53.5 | 62.0 | 58.9 | 66.0 | **68.9** |
| OLMoE-1B-7B | Reranking | 41.7 | 51.3 | 51.0 | 53.2 | **55.1** |
| | STS | 49.4 | 63.5 | 44.2 | 67.8 | **71.1** |
| | Summarization | 25.6 | 28.9 | 29.7 | 30.4 | **30.5** |
| | **Average** | 38.3 | 48.0 | 44.5 | 48.7 | **53.2** |
| | Classification | 48.8 | 58.0 | 35.2 | 54.0 | **59.7** |
| | Clustering | 14.3 | 34.2 | 29.2 | 30.1 | **37.5** |
| | PairClassification | 51.9 | 60.5 | 56.0 | 60.3 | **66.6** |
| Qwen1.5-MoE-A2.7B | Reranking | 41.0 | 46.6 | 45.0 | 51.1 | **56.8** |
| | STS | 48.3 | 50.1 | 50.0 | 64.3 | **69.0** |
| | Summarization | 27.0 | 23.0 | 21.9 | 27.3 | **31.0** |
| | **Average** | 38.6 | 45.4 | 39.6 | 47.9 | **53.4** |
| | Classification | 48.6 | 56.4 | 55.4 | 53.0 | **60.4** |
| | Clustering | 17.8 | 29.0 | 20.3 | 28.5 | **32.8** |
| | PairClassification | 57.4 | 59.8 | 53.8 | 63.3 | **67.9** |
| DeepSeekMoE-16B | Reranking | 43.8 | 45.7 | 40.9 | 50.6 | **52.4** |
| | STS | 52.8 | 49.0 | 37.1 | 63.4 | **68.1** |
| | Summarization | 29.1 | 24.4 | 25.7 | 29.2 | **30.7** |
| | **Average** | 41.6 | 44.0 | 38.9 | 48.0 | **52.0** |

*Table 2.* Performance comparison of USMoE, Token Choice (TC), Expert Choice (EC), and MoEE across across MTEB Tasks with **PromptEOL (Jiang et al., 2024b)**. The best result for each row is highlighted in **bold**.

103, and One Billion Words) for 100k iterations, following the implementation in (Chen et al., 2023). Then we fine-tune the pre-trained weights for text classification tasks, including SST-2 (Socher et al., 2013), SST-5 (Socher et al., 2013), IMDB (Maas et al., 2011), and BANKING77 (Casanueva et al., 2020). More implementation details and additional results are provided in the Appendix A.

**Key Results.** As shown in Table 3, USMoE demonstrates consistent improvements over TC and EC baselines across all pre-training datasets. The robustness of our approach is highlighted by its sustained performance in corrupted settings, where it maintains a significant lead over traditional SMoE models. A distinctive feature of USMoE is the flexibility of its fractional TopK selection; it achieves performance parity with, or surpasses, integer-based models while utilizing fewer experts. Specifically, on enwik8 and text8, USMoE with $k = 1.5$ outperforms TC with $k = 2$. This architectural flexibility enables a **14% reduction in FLOPs** relative to standard SMoE configurations, facilitating deployment in IoT and edge computing contexts. We provide extended results for 400M parameter models and downstream tasks in Section A.

### 4.2.2. VISION

**Effiency.** To ensure a thorough performance comparison, we evaluate against two categories of baselines that are closely aligned with USMoE, including Token Choice, Expert Choice, and SoftMoE (Puigcerver et al., 2024), currently among the most advanced MoE models in the vision domain. We explore two Mixture of Experts configurations for Vision Transformer (Dosovitskiy et al., 2021): (1) a small model with 10 million parameters (10M), and (2) a

| Setting | Dataset | USMoE | | TC | EC |
|---|---|---|---|---|---|
| | | $k = 2$ | $k = 1.5$ | $k = 2$ | $k = 2$ |
| Original | Enwik8 ($\downarrow$) | **1.18** | 1.19 | 1.20 | **1.18** |
| | Text8 ($\downarrow$) | **1.20** | 1.28 | 1.29 | 1.24 |
| | WikiText-103 ($\downarrow$) | **29.20** | 30.67 | 30.16 | 29.83 |
| | lm1b ($\downarrow$) | **56.90** | 57.55 | 58.00 | 58.60 |
| Corrupt | Enwik8 ($\downarrow$) | **1.75** | 1.78 | 1.77 | 1.76 |
| | Text8 ($\downarrow$) | **1.83** | 1.95 | 1.86 | 1.89 |
| | WikiText-103 ($\downarrow$) | **38.45** | 40.39 | 39.31 | 39.28 |
| | lm1b ($\downarrow$) | **68.43** | 70.47 | 79.73 | 71.75 |

*Table 3.* Performance comparison of USMoE, Token Choice (TC), and Expert Choice (EC) across multiple datasets for Transformer-XL(20M) (Dai et al., 2019), with BPC on the Enwik8 and Text8 test sets, and perplexity on the WikiText-103 and One Billion Word test sets. Lower values are better, with the best results highlighted in **bold**.

| Size | Dataset | USMoE | TC | EC | SoftMoE |
|---|---|---|---|---|---|
| 10M | Cifar10 | **89.6**$_{\pm 0.3}$ | 88.7$_{\pm 0.2}$ | 88.9$_{\pm 0.3}$ | 85.6$_{\pm 0.3}$ |
| | Cifar100 | **66.6**$_{\pm 0.5}$ | 65.4$_{\pm 0.5}$ | 65.7$_{\pm 0.4}$ | 61.4$_{\pm 0.3}$ |
| | STL-10 | **66.7**$_{\pm 0.4}$ | 66.4$_{\pm 0.1}$ | 66.1$_{\pm 0.4}$ | 65.4$_{\pm 0.2}$ |
| | SVHN | **95.6**$_{\pm 0.1}$ | 95.1$_{\pm 0.1}$ | 95.0$_{\pm 0.1}$ | 94.8$_{\pm 0.1}$ |
| | ImageNet-1K | **60.2**$_{\pm 0.1}$ | 56.6$_{\pm 0.5}$ | 56.2$_{\pm 0.4}$ | 46.8$_{\pm 0.6}$ |
| 110M | Cifar10 | **91.5**$_{\pm 0.5}$ | 85.7$_{\pm 0.8}$ | 87.4$_{\pm 0.7}$ | 80.3$_{\pm 1.0}$ |
| | Cifar100 | **67.3**$_{\pm 0.5}$ | 55.5$_{\pm 2.8}$ | 66.2$_{\pm 0.9}$ | 42.9$_{\pm 1.4}$ |
| | STL-10 | **66.2**$_{\pm 0.1}$ | 64.4$_{\pm 0.2}$ | 65.5$_{\pm 0.4}$ | 63.9$_{\pm 1.2}$ |
| | SVHN | **96.1**$_{\pm 0.1}$ | 94.5$_{\pm 0.4}$ | 93.2$_{\pm 0.2}$ | 93.5$_{\pm 0.1}$ |
| | ImageNet-1K | **73.5**$_{\pm 0.4}$ | 72.0$_{\pm 0.4}$ | 70.9$_{\pm 0.5}$ | 71.2$_{\pm 0.3}$ |
| **Avg.** | All | **77.3**$_{\pm 0.3}$ | 74.4$_{\pm 1.5}$ | 75.5$_{\pm 1.2}$ | 70.6$_{\pm 1.5}$ |

*Table 4.* Accuracy of VIT-MoE evaluated on vision classification tasks. Each method is evaluated 3 times, reporting the mean and standard deviation. Higher is better, the best results are in bold.

large model with 110 million parameters (110M). As shown in Table 4, USMoE consistently outperforms Vision Transformer variants with Token Choice, Expert Choice, and SoftMoE across all eight tasks and four image classification datasets. Our experiments are conducted three times on four datasets (CIFAR-10, CIFAR-100, STL-10, SVHN, and ImageNet1K), using different random seeds. We report average results along with standard deviations. Across diverse vision datasets, USMoE consistently delivers the highest mean accuracy with significantly lower variance compared to traditional baselines like Token Choice and SoftMoE. With an overall average score of **77.3%**, USMoE proves to be a more robust and reliable routing mechanism for both specialized and general purpose vision tasks.

### 4.3. Explainability of the Unified Perspective

In this section, we discuss the application of the Unified Perspective to explainability in Mixture of Experts models. The Unified Perspective enables us to treat experts and tokens symmetrically, allowing their relationships to

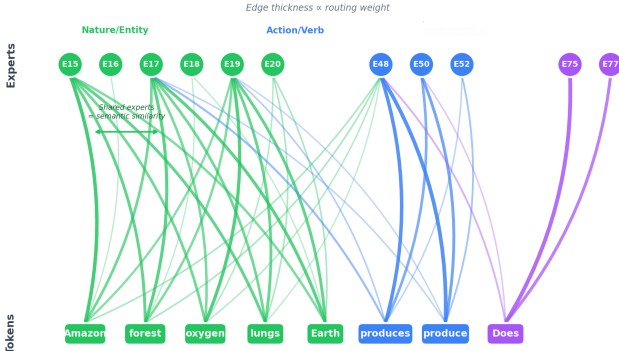

*Figure 5.* Explainability comparison between USMoE and the conventional view on the BoolQ dataset using Qwen3-30B-A3B-Instruct. Under USMoE, tokens routed to the same experts exhibit coherent semantic characteristics, while experts that attend to similar token sets tend to correspond to interpretable sub-domains (e.g., "Nature" experts spanning Expert 15 to Expert 20). This unified perspective reveals structured token–expert relationships that are less apparent under conventional routing formulations. Best viewed in color.

be represented as a bipartite graph where nodes correspond to tokens and experts, and edges are weighted by router scores. This formulation allows us to understand expert relationships through their shared token connections, and conversely, to analyze token relationships through their co-activation of experts. For example, in Figure 5, certain experts are predominantly connected to tokens associated with nature-related topics, suggesting shared specialization in this domain, while other experts exhibit stronger connections to action-oriented tokens. This graph-based view provides a principled framework for understanding and explaining the behavior of MoE-based LLMs.

### 4.4. Complexity Discussion

We analyze the computational complexity of USMoE relative to standard Token Choice routing, used in conventional SMoE layers. Let $B$, $L$, $D$, and $N$ denote the batch size, sequence length, hidden dimension, and number of experts, respectively, and let $k$ be the number of selected experts per token. The routing cost can be decomposed into two parts: (i) computing token–expert compatibility scores and (ii) selecting the active token–expert assignments.

For Token Choice routing, the router first computes scores for all $BL$ tokens over $N$ experts, which costs

$$O(BLDN + BLN),$$

where the additional $O(BLN)$ term accounts for normalization or score transformation. It then performs a Top-$k$ selection over $N$ experts independently for each token, giving a selection cost of

$$O(BL \cdot \text{TopK}(N, k)).$$

Thus, the overall complexity of Token Choice routing is

$$O(BLDN + BLN) + O(BL \cdot \text{TopK}(N, k)).$$

USMoE uses the same token–expert score computation, and therefore has the same score-computation complexity:

$$O(BLDN + BLN).$$

The difference lies only in the selection step. Instead of performing $L$ independent token-wise Top-$k$ operations, USMoE performs a global Top-$Lk$ selection over the $LN$ token–expert pairs for each batch element. Its selection cost is therefore

$$O(B \cdot \text{TopK}(LN, Lk)),$$

and the total routing complexity becomes

$$O(BLDN + BLN) + O(B \cdot \text{TopK}(LN, Lk)).$$

Under standard near-linear Top-$K$ implementations, the selection term is lower order compared to the dense score computation term $O(BLDN)$, especially when the hidden dimension $D$ is large. Therefore, USMoE and Token Choice routing share the same asymptotic complexity dominated by score computation. Importantly, USMoE changes the selection geometry from local token-wise assignment to global token–expert assignment, but it does not introduce an additional asymptotic routing cost over conventional SMoE.

### 4.5. Ablation Studies

We investigate the effectiveness and robustness of USMoE to the different hyper-parameter settings.

#### 4.5.1. UNIFIED LEARNING STRATEGY COMPARISON

The Unified Mechanism is implemented using two approaches: (1) a sequence-based method that compares all tokens within a sequence (referred to as "USMoE-Sequence") and (2) a batch-based method that compares all tokens within a batch or mini-batch (referred to as "USMoE-Batch"). We evaluate both approaches on the Classification task, with results presented in Table 5. The findings indicate that both methods outperform the Expert Choice and Token Choice approaches, demonstrating the effectiveness of our method. Notably, the sequence-based approach achieves superior performance in the Classification task, as it ensures that no important tokens are missed within a sequence - an assurance that the batch/mini-batch implementation may not always provide.

#### 4.5.2. ROBUSTNESS TO THE CONTROLLING FACTOR

The Unified Score ($\alpha$) enables the model to adjust its scoring mechanism, either favoring a diverse set of experts per

sequence or distributing experts more evenly across tokens. We evaluate the robustness of the controlling factor $\alpha$ on the classification task using the *DeepSeekMoE-16B* model, with results presented in Table 6. When $\alpha = 0.0$, the scoring mechanism aligns with Token Choice, while at $\alpha = 1.0$, it follows Expert Choice.

Overall, USMoE demonstrates strong effectiveness within the range of $\alpha \in (0.3, 0.7)$, striking a balance between expert diversity and token importance. This range provides an optimal trade-off between enforcing SMoE's unified policy and enhancing traditional approaches for the task, leading to superior overall performance. Notably, all tested $\alpha$ configurations outperform the Expert Choice approach ($\alpha = 1.0$).

| Model | Dataset | TC | EC | USMoE-Sequence | USMoE-Batch |
|---|---|---|---|---|---|
| | Emotion | 27.4 | 26.5 | **27.8** | 27.4 |
| DeepSeekMoE-16B | Toxic | **60.4** | 58.1 | 60.1 | 59.2 |
| | Tweet | 51.9 | 49.5 | **52.5** | 51.7 |

*Table 5.* Comparison of USMoE, Token Choice (TC), and Expert Choice (EC) on the classification task. Higher values are better, with the best results highlighted in **bold**.

| Model | Dataset | $\alpha$ | | | | | |
|---|---|---|---|---|---|---|---|
| | | 0.0 | 0.3 | 0.5 | 0.7 | 0.9 | 1.0 |
| | Emotion | 27.4 | 27.1 | **27.8** | 27.6 | 27.7 | 26.5 |
| DeepSeekMoE-16B | Toxic | **60.4** | 60.0 | 60.1 | 56.8 | 57.3 | 58.1 |
| | Tweet | 51.9 | **53.2** | 52.5 | 53.3 | 52.9 | 49.5 |

*Table 6.* Performance comparison of DeepSeekMoE-16B across different classification datasets with varying $\alpha$ values. Higher is better; best results are in **bold**.

### 4.5.3. CONTRIBUTION ANALYSIS

We ablate the two components of USMoE: the Unified Mechanism (UM), which performs global token–expert allocation, and the Unified Score (US), which combines token-choice and expert-choice signals. Table 7 shows that both components contribute consistently across tasks. On average, UM accounts for the majority of the gain over the original TC baseline, contributing $60\%$, while US provides the remaining complementary improvement of $40\%$. This indicates that global allocation is the primary driver of USMoE's improvement, while the unified scoring function further strengthens performance.

## 5. Conclusion

In this work, we reinterpret Token Choice and Expert Choice Sparse Mixture of Experts (SMoE) through a linear programming lens, revealing their inherent limitations. Based on this analysis, we propose Unified Sparse Mixture of Experts (USMoE) - a novel and efficient framework that advances SMoE by introducing a unified mechanism and unified score learning. We theoretically prove that USMoE achieves supe-

| Task | Original (TC) | UM Only | USMoE | UM Contrib. | US Contrib. |
|---|---|---|---|---|---|
| ARC-C | 0.631 | 0.645 | **0.646** | 0.96 | 0.04 |
| ARC-E | 0.838 | 0.844 | **0.846** | 0.74 | 0.26 |
| BoolQ | 0.886 | 0.893 | **0.894** | 0.87 | 0.13 |
| OBQA | 0.454 | 0.458 | **0.460** | 0.58 | 0.42 |
| PIQA | 0.805 | 0.807 | **0.810** | 0.34 | 0.66 |
| WinoGrande | 0.733 | 0.733 | **0.736** | 0.11 | 0.89 |
| Average | 0.724 | 0.730 | **0.732** | 0.60 | 0.40 |

*Table 7.* Ablation of the Unified Mechanism (UM) and Unified Score (US) on Qwen3-30B-A3B-Instruct under zero-shot evaluation. Contributions are measured relative to the total improvement from TC to USMoE.

rior expert selection compared to traditional methods, effectively improving expert learning capacity while mitigating expert collapse. As a result, USMoE learns more robust expert representations and overcomes the representation collapse issues commonly observed in conventional SMoE training. Extensive experiments across diverse domains, including vision and large language models (LLMs), under both clean and adversarial settings (covering training-free and training scenarios), demonstrate that USMoE enables more efficient and effective training and inference compared to the advanced MoE-based LLMs.

## Limitations

This work studies USMoE as a unified routing framework for LLMs, covering training-free adaptation, finetuning, and training-from-scratch settings. Although our results show consistent gains across multiple benchmarks and routing configurations, several limitations remain. First, due to computational constraints, our experiments are limited to medium-scale evaluation settings and models up to Qwen3-30B-A3B. Evaluating USMoE on substantially larger MoE models, especially those beyond 100B parameters, remains an important direction for future work. Second, while we compare against representative Token-Choice, Expert-Choice, and advanced MoE routing baselines, broader comparisons with recent large reasoning-oriented MoE models, such as DeepSeek-R1-style systems, would further clarify the scalability and generality of the proposed routing perspective. Finally, although USMoE improves the efficiency-accuracy trade-off in our experiments, its practical deployment may require optimized kernels for global token-expert selection to fully realize the computational benefits at large batch sizes and long context lengths. We leave these large-scale evaluations, stronger systems-level optimizations, and extensions to broader reasoning workloads for future work.

## Impact Statement

This research contributes to the technical advancement of Machine Learning. Our experiments were conducted in an academic setting using public benchmarks, without the

use of human subjects or proprietary data. While this work focuses on foundational methodology, we recognize that Large Language Models (LLMs) trained on web-scale data can inherit and amplify societal biases, including those related to gender and race. Furthermore, we acknowledge the significant computational footprint and environmental costs associated with training large-scale models. We view the mitigation of these biases and the optimization of resource efficiency as essential ongoing challenges for the field.

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

# A. Appendix

# Supplementary Material for "Rethinking Mixture of Experts from a Unified Perspective"

This supplementary material provides additional theoretical, algorithmic, and empirical details to support the main paper. Appendix A.1 presents the theoretical proofs underlying the unified formulation introduced in Section 3. Appendix A.2 discusses additional related work and further positions USMoE with respect to existing Token-Choice, Expert-Choice, and advanced MoE routing methods. Appendix A.3 gives the complete USMoE algorithm, including the construction of unified routing scores and the global token–expert selection procedure. Appendix A.4 provides both theoretical analysis and empirical evidence showing how USMoE mitigates the token-dropping issue commonly observed in Expert-Choice MoEs. Appendix A.5 reports additional experimental results, including extended benchmark comparisons, efficiency analysis, and ablation studies. Appendix A.6 provides a deeper analysis of Token-Choice and Expert-Choice routing, explaining their limitations and why the unified routing perspective leads to stronger performance. Finally, Appendix A.7 describes implementation details, including model configurations, evaluation protocols, and hyperparameter settings.

## A.1. Theoretical Proof

### A.1.1. THEORETICAL PROOF FOR SECTION 3

**Definition A.1** (General Routing Function $\mathcal{R}$). Let $S \in \mathbb{R}^{T \times N}$ be the compatibility score matrix between $T$ tokens and $N$ experts, and let $c \in \mathbb{N}$ denote a global routing budget that constrains the total number of token-to-expert assignments. Let $\text{TopK}_R$, $\text{TopK}_C$, and $\text{TopK}$ denote the TopK selection operators applied row-wise, column-wise, and globally, respectively. We define a general routing function $\mathcal{R}(S, c, \texttt{mode})$ that constructs a sparse binary routing matrix $X \in \{0, 1\}^{T \times N}$, where $x_{ij} = 1$ indicates that token $i$ is routed to expert $j$, as follows:

$$
X = \mathcal{R}(S, c, \texttt{mode}) = \begin{cases} \text{TopK}(S, c), & \texttt{mode} = \text{USMoE}, \\ \text{TopK}_R(S, \lfloor \frac{c}{T} \rfloor), & \texttt{mode} = \text{Token Choice}, \\ \text{TopK}_C(S, \lfloor \frac{c}{N} \rfloor), & \texttt{mode} = \text{Expert Choice}, \end{cases}
$$

**Lemma A.2.** *The Unified Mechanism (USMoE), as defined in Definition A.1.1, yields a superior solution to the optimization problem in Equation 2 compared to both Token Choice and Expert Choice mechanisms.*

*Proof.* Let the ordered arrays $(S_1^t, S_2^t, \ldots, S_T^t)$, $(S_1^e, S_2^e, \ldots, S_T^e)$, and $(S_1^u, S_2^u, \ldots, S_T^u)$ represent the top-$T$ compatibility scores selected by the *Token Choice*, *Expert Choice*, and *Unified Score* mechanisms, respectively. These scores are sorted in ascending order such that

$$S_1^t \leq S_2^t \leq \cdots \leq S_T^t, \quad S_1^e \leq S_2^e \leq \cdots \leq S_T^e, \quad S_1^u \leq S_2^u \leq \cdots \leq S_T^u.$$

We claim the following inequality holds:
$$
\begin{aligned}
S_i^t &\leq S_i^u, \quad \forall i \in [1, T], \\
S_i^e &\leq S_i^u, \quad \forall i \in [1, T].
\end{aligned} \tag{8}
$$

Let $x \in \mathbb{R}^{T \times d}$ denote a matrix of token embeddings and $\mathcal{E} \in \mathbb{R}^{d \times N}$ the matrix of expert embeddings, where $d$ is the embedding dimension, $T$ is the number of tokens, and $N$ is the number of experts. The similarity matrix $S \in \mathbb{R}^{T \times N}$ between tokens and experts is computed via a dot product:

$$S = x \cdot \mathcal{E}. \tag{9}$$

Since the softmax function is monotonic, the top-$k$ selection remains invariant under softmax transformation:

$$\text{TopK}(\text{softmax}(S), k) = \text{TopK}(S, k). \tag{10}$$

For the Token Choice approach, the top similarity score for each token is defined as:

$$S_j^t = \max_{k \in [1,N]} S_{jk}, \tag{11}$$

and across all tokens, the top-$T$ scores are aggregated into $\{S_i^t\}_{i=1}^T$.

In contrast, the Unified Mechanism selects the top-$T$ scores globally across all token-expert pairs:

$$S_j^u = \max_{(q,k) \in [1,T] \times [1,N]} S_{qk}. \tag{12}$$

By construction, it follows that:

$$S_i^t \leq S_i^u, \quad \forall i \in [1,T], \tag{13}$$

since Token Choice selects top experts per token independently, while Unified Score considers all token-expert pairs jointly.

Similarly, in the Expert Choice approach, each expert selects its most compatible token:

$$S_i^e = \max_{q \in [1,T]} S_{qi}, \quad \forall i \in [1,N], \tag{14}$$

and again, the top-$T$ scores across experts are extracted as $\{S_i^e\}_{i=1}^T$. As the Unified approach considers the global maximums, we have:

$$S_i^e \leq S_i^u, \quad \forall i \in [1,T]. \tag{15}$$

Combining (13) and (15), we conclude:

$$S_i^t \leq S_i^u, \quad S_i^e \leq S_i^u, \quad \forall i \in [1,T], \tag{16}$$

which completes the proof. □ □

### A.1.2. PROOF OF LEMMA 3.7

*Proof.* We follow prior analyses of representation collapse in sparse Mixture-of-Experts models through the Jacobian of the MoE layer with respect to the input representation (Chi et al., 2022; Do et al., 2023). Let $x \in \mathbb{R}^d$ denote an input token representation, let $N$ be the number of experts, and let $E_i(x)$ denote the output of expert $i$. For clarity, we present the argument for a single token; the batched and sequence-level case follows by applying the same argument independently across token positions.

For a standard Token Choice SMoE layer, the output can be written as

$$y(x) = \sum_{i \in \mathcal{K}(x)} S_i(x) E_i(x), \tag{17}$$

where $S_i(x)$ is the router score and $\mathcal{K}(x)$ is the selected Top-$K$ expert set. Ignoring the non-differentiability of the discrete Top-$K$ boundary and considering the local region where $\mathcal{K}(x)$ is fixed, the Jacobian takes the form

$$J_{\text{SMoE}}(x) = \sum_{i \in \mathcal{K}(x)} S_i(x) J_{E_i}(x) + \sum_{i \in \mathcal{K}(x)} E_i(x) \nabla_x S_i(x)^\top. \tag{18}$$

The first term captures the contribution of the selected experts themselves, while the second term captures the contribution of the gating function.

When the router is implemented by a softmax over token–expert logits, we have

$$\nabla_x S_i(x) = S_i(x) \sum_{j=1}^N (\delta_{ij} - S_j(x)) \nabla_x r_j(x), \tag{19}$$

where $r_j(x)$ is the logit for expert $j$. If the router logits are linear in $x$, i.e., $r_j(x) = e_j^\top x$, then $\nabla_x r_j(x) = e_j$, and the gating part of the Jacobian can be written as

$$\sum_{j=1}^{N} c_j(x) e_j^\top, \tag{20}$$

for some coefficient vectors $c_j(x)$ depending on the expert outputs and router scores. Therefore, the gating-induced component of the Jacobian lies in the span of the router expert embeddings $\{e_j\}_{j=1}^{N}$. Since typically $N \ll d$, this component can be low-rank relative to the representation dimension. This provides a Jacobian-based view of representation collapse: the routing gradient is constrained to a limited expert-embedding subspace.

USMoE modifies this structure by combining token-choice and expert-choice routing signals. Let

$$S_U(x) = (1 - \alpha)S_t(x) + \alpha S_e(x), \qquad \alpha \in [0, 1], \tag{21}$$

where $S_t(x)$ denotes the token-choice routing score and $S_e(x)$ denotes the expert-choice routing score. The USMoE output is

$$y_U(x) = \sum_{i \in \mathcal{K}_U(x)} S_{U,i}(x) E_i(x). \tag{22}$$

Again considering a local region where the selected set is fixed, its Jacobian is

$$J_U(x) = \sum_{i \in \mathcal{K}_U(x)} S_{U,i}(x) J_{E_i}(x) + \sum_{i \in \mathcal{K}_U(x)} E_i(x) \nabla_x S_{U,i}(x)^\top. \tag{23}$$

Since $S_U(x)$ is a mixture of the two routing signals, we have

$$\nabla_x S_{U,i}(x) = (1 - \alpha)\nabla_x S_{t,i}(x) + \alpha\nabla_x S_{e,i}(x). \tag{24}$$

Substituting this into Eq. (23) gives

$$
J_U(x) = \underbrace{\sum_{i \in \mathcal{K}_U(x)} S_{U,i}(x) J_{E_i}(x)}_{\text{expert-output component}}
$$
$$
+ (1 - \alpha) \underbrace{\sum_{i \in \mathcal{K}_U(x)} E_i(x)\nabla_x S_{t,i}(x)^\top}_{\text{token-choice gating component}}
$$
$$
+ \alpha \underbrace{\sum_{i \in \mathcal{K}_U(x)} E_i(x)\nabla_x S_{e,i}(x)^\top}_{\text{expert-choice gating component}}. \tag{25}
$$

Thus, unlike standard SMoE, whose gating-induced Jacobian is determined by a single routing normalization, USMoE contains two gating components induced by different normalization mechanisms. Equivalently, the gating part can be written as

$$J_U^{\text{gate}}(x) = (1 - \alpha)J_t^{\text{gate}}(x) + \alpha J_e^{\text{gate}}(x), \tag{26}$$

where

$$J_t^{\text{gate}}(x) = \sum_{j=1}^{N} c_j(x) e_j^\top, \qquad J_e^{\text{gate}}(x) = \sum_{j=1}^{N} d_j(x) e_j^\top. \tag{27}$$

Here, $c_j(x)$ and $d_j(x)$ are coefficient vectors induced by token-choice and expert-choice routing, respectively.

The important distinction is not merely that Eq. (26) contains more summands, since rank depends on linear independence rather than the number of terms. Rather, the token-choice and expert-choice components are produced by different normalizations and therefore induce different coefficient structures. When $S_t$ and $S_e$ are not perfectly correlated, the corresponding coefficient vectors $\{c_j(x)\}$ and $\{d_j(x)\}$ are not collinear in general. Under this non-degeneracy condition, the sum of the two components spans a larger effective subspace than either component alone:

$$\text{rank}_{\text{eff}}\left(J_U^{\text{gate}}(x)\right) \geq \max\left\{\text{rank}_{\text{eff}}\left(J_t^{\text{gate}}(x)\right), \text{rank}_{\text{eff}}\left(J_e^{\text{gate}}(x)\right)\right\}, \tag{28}$$

with a strict increase whenever the two gating-induced subspaces contribute non-overlapping directions.

Therefore, USMoE mitigates representation collapse by enriching the Jacobian structure of the MoE layer. Instead of relying on a single token-wise routing signal, USMoE combines token-choice and expert-choice signals, producing a routing Jacobian with higher effective rank under mild non-degeneracy assumptions. This supports Lemma 3.7. □

### A.1.3. PROOF OF LEMMA 3.5: ROBUSTNESS TO NOISY TOKENS

*Proof.* **Setup.** Let $S^* \in \mathbb{R}^{T \times N}$ denote the clean compatibility score matrix. For noisy tokens $i \in \mathcal{N}$, the observed scores are:

$$S_{ij} = S_{ij}^* + \epsilon_{ij}, \quad \text{where } \|\epsilon_i\|_\infty \leq \delta.$$

For clean tokens $i \notin \mathcal{N}$, we have $S_{i,:} = S_{i,:}^*$.

Let $j_i^* = \arg\max_j S_{ij}^*$ denote the correct expert assignment for token $i$, and define the score margin:

$$\Delta_{\min}^{(i)} = \min_{j \neq j_i^*} \left( S_{ij_i^*}^* - S_{ij}^* \right) > 0.$$

#### Part 1: Misrouting under Token Choice.

Under Token Choice, each token $i$ independently selects its top-$k$ experts based on row-wise scores. Consider top-1 routing for simplicity. Token $i$ is misrouted to expert $j \neq j_i^*$ if:

$$S_{ij} > S_{ij_i^*} \quad \Longleftrightarrow \quad S_{ij}^* + \epsilon_{ij} > S_{ij_i^*}^* + \epsilon_{ij_i^*}.$$

Rearranging:

$$\epsilon_{ij} - \epsilon_{ij_i^*} > S_{ij_i^*}^* - S_{ij}^* = \Delta_{ij}^*.$$

Since $\|\epsilon_i\|_\infty \leq \delta$, we have $\epsilon_{ij} - \epsilon_{ij_i^*} \leq 2\delta$. Therefore, misrouting is possible whenever:

$$\Delta_{ij}^* < 2\delta \quad \text{for some } j \neq j_i^*.$$

Equivalently, token $i$ is vulnerable to misrouting under Token Choice if $\Delta_{\min}^{(i)} < 2\delta$.

The key observation is that Token Choice routing depends *only* on the relative scores within row $i$. The noise $\epsilon_i$ directly perturbs this local comparison, and no information from other tokens can mitigate its effect.

#### Part 2: Robustness of the Unified Mechanism (USMoE).

Under USMoE, routing is determined by selecting the top-$c$ entries from the *entire* score matrix $S \in \mathbb{R}^{T \times N}$ globally (Equation 6). A token-expert pair $(i, j)$ is selected only if:

$$S_{ij} \in \text{Top-}c\big(\{S_{i'j'}\}_{i'=1,j'=1}^{T,N}\big).$$

For a noisy token $i \in \mathcal{N}$ to be misrouted to expert $j \neq j_i^*$, two conditions must hold:

(C1) The corrupted score $S_{ij} = S_{ij}^* + \epsilon_{ij}$ must be large enough to enter the global top-$c$.

(C2) The correct pair $(i, j_i^*)$ must either fail to enter the top-$c$, or the incorrect pair $(i, j)$ must be selected instead due to capacity constraints.

**Global Competition Effect.** Unlike Token Choice, where selection depends only on row $i$, USMoE requires the noisy score $S_{ij}$ to compete against scores from *all other tokens*, including the $T - |\mathcal{N}|$ clean tokens.

Let $\tau_c$ denote the $c$-th largest score in the matrix $S$. For $(i, j)$ to be selected:

$$S_{ij}^* + \epsilon_{ij} \geq \tau_c.$$

For clean tokens $i' \notin \mathcal{N}$, their scores $S^*_{i'j'}$ are unperturbed. If the clean score matrix $S^*$ contains many high-scoring entries, the threshold $\tau_c$ remains high, making it difficult for noise-elevated scores to qualify.

Formally, define the *clean competition margin*:

$$\gamma = \tau_c^* - \max_{i \in \mathcal{N}, j \neq j_i^*} S^*_{ij},$$

where $\tau_c^*$ is the $c$-th largest entry in the clean matrix $S^*$. If $\gamma > \delta$, then even with maximal noise, incorrect pairs from noisy tokens cannot enter the top-$c$:

$$S^*_{ij} + \epsilon_{ij} \leq S^*_{ij} + \delta < \tau_c^* \leq \tau_c.$$

Thus, misrouting is prevented whenever clean tokens provide sufficient competition.

**Part 3: Comparing Misrouting Probabilities.**

We now show $P_{\text{mis}}^{\text{USMoE}} \leq P_{\text{mis}}^{\text{TC}}$.

Under Token Choice, token $i \in \mathcal{N}$ is misrouted if:

$$A_i^{\text{TC}} : \quad \exists j \neq j_i^* \text{ such that } \epsilon_{ij} - \epsilon_{ij_i^*} > \Delta_{ij}^*.$$

Under USMoE, token $i$ is misrouted only if:

$$A_i^{\text{USMoE}} : \quad A_i^{\text{TC}} \wedge \left(S^*_{ij} + \epsilon_{ij} \geq \tau_c\right).$$

Since $A_i^{\text{USMoE}} \subseteq A_i^{\text{TC}}$ (misrouting under USMoE requires misrouting under the local Token Choice criterion *and* passing the global threshold), we have:

$$P_{\text{mis}}^{\text{USMoE}}(i) = \Pr[A_i^{\text{USMoE}}] \leq \Pr[A_i^{\text{TC}}] = P_{\text{mis}}^{\text{TC}}(i).$$

Averaging over all noisy tokens:

$$P_{\text{mis}}^{\text{USMoE}} = \frac{1}{|\mathcal{N}|} \sum_{i \in \mathcal{N}} P_{\text{mis}}^{\text{USMoE}}(i) \leq \frac{1}{|\mathcal{N}|} \sum_{i \in \mathcal{N}} P_{\text{mis}}^{\text{TC}}(i) = P_{\text{mis}}^{\text{TC}}.$$

**Strictness of the Inequality.** The inequality is strict when $\Pr[S^*_{ij} + \epsilon_{ij} \geq \tau_c \mid A_i^{\text{TC}}] < 1$, which occurs when:

- The number of clean tokens $T - |\mathcal{N}|$ is sufficiently large, or

- The clean scores $S^*_{i'j'}$ for $i' \notin \mathcal{N}$ are sufficiently high to maintain a large $\tau_c$.

In typical settings where noisy tokens constitute a small fraction of the total ($|\mathcal{N}| \ll T$), the global competition from clean tokens provides substantial protection against misrouting.

This completes the proof. □

A.1.4. PROOF OF PROPOSITION 3.3

*Proof.* Let $S \in \mathbb{R}^{t \times n}$ be the similarity matrix, and let $c \in \mathbb{N}$ be a global routing budget. The goal is to find a binary matrix $X \in \{0, 1\}^{t \times n}$ such that

$$\sum_{i=1}^{t} \sum_{j=1}^{n} x_{ij} \leq c$$

and the total score

$$M = \langle S, X \rangle = \sum_{i=1}^{t} \sum_{j=1}^{n} S_{ij} x_{ij}$$

is maximized.

Let $X_{\text{USMoE}} = \text{TopK}(S, c)$ denote the binary matrix that marks the positions of the top-$c$ largest entries in $S$. By construction, $X_{\text{USMoE}}$ sets exactly $c$ entries of $S$ with the highest values to 1 and all others to 0.

Suppose there exists another feasible solution $X_T \in \{0, 1\}^{t \times n}$ with

$$\sum_{i,j} (X_T)_{ij} \leq c$$

and assume for contradiction that

$$\langle S, X_T \rangle > \langle S, X_{\text{USMoE}} \rangle.$$

Since $X_T$ selects at most $c$ entries from $S$, and the values of the selected entries are summed to compute $\langle S, X_T \rangle$, the only way for $\langle S, X_T \rangle$ to exceed $\langle S, X_{\text{USMoE}} \rangle$ is if some entries selected by $X_T$ are larger than those selected by $X_{\text{USMoE}}$. However, this contradicts the definition of $X_{\text{USMoE}}$ as containing the top-$c$ largest values in $S$.

Therefore, for any feasible $X_T$, we must have

$$\langle S, X_T \rangle \leq \langle S, X_{\text{USMoE}} \rangle,$$

which proves that $X_{\text{USMoE}}$ is the optimal solution.

This completes the proof. $\qquad\square$

### A.2. Related Work (Cont.)

Token Choice treats all tokens equally, which has raised concerns among researchers (Wu et al., 2021; Hou et al., 2022; Lin et al., 2025), while Expert Choice suffers from token-dropping issues. Additionally, SMoE faces the challenge of representation collapse, where experts produce similar outputs. Various solutions have been proposed, such as XMoE, which employs low-dimensional routing scores (Chi et al., 2022), and SMoE-dropout, which gradually activates more experts (Chen et al., 2023). Other approaches, including HyperRouter (Do et al., 2023) and StableMoE (Dai et al., 2022), focus on enhancing router stability and robustness. Although these advancements have improved SMoE models, representation collapse remains a persistent issue (Pham et al., 2024; Do et al., 2025a). Our approach addresses this by optimizing the alignment between tokens and the most suitable experts, expanding expert specialization and mitigating collapse.

The most relevant work to ours is *Expert Race*(Yuan et al., 2025), which proposes a flexible routing strategy for diffusion transformers. However, similar to the traditional Expert Choice (EC) approach, it suffers from a critical limitation, *information leakage* - where a model unintentionally accesses future tokens (Raposo et al., 2024; Wang et al., 2024), which becomes especially problematic in autoregressive language modeling settings(Raposo et al., 2024; Wang et al., 2024). In contrast, we propose a modified version of Expert Choice (Zhou et al., 2022) by replacing the *softmax* mapping function with a *sigmoid* function and limiting comparisons to within the token dimension. This design effectively mitigates information leakage, making Expert Choice more suitable for large language models.

Additionally, we introduce two novel modules: (1) the Unified Score, a linear combination of the mapping functions from Token Choice and Expert Choice, and (2) the Unified Mechanism, which jointly considers both expert and token dimensions when selecting experts. Both theoretical analysis and empirical results demonstrate that our method outperforms traditional SMoE approaches in terms of both performance and robustness.

### A.3. USMoE algorithm

### A.4. How Does USMoE Mitigate Token Dropping in Expert-Choice MoEs?

Expert-Choice (EC) routing improves expert-side load balancing by allowing each expert to select its preferred tokens. However, because selection is performed from the expert perspective, EC does not guarantee that every token is assigned to at least one expert. As a result, some tokens may be dropped entirely, causing potential information loss during sparse feed-forward computation. In contrast, Token-Choice (TC) routing avoids token dropping by construction, since each token independently selects its top-$K$ experts. USMoE preserves this token-coverage property in practice while benefiting from a global selection mechanism over token–expert pairs.

---

**Algorithm 1** Unified Sparse Mixture-of-Experts (USMoE) Layer

---

1: **Input:** Input tensor $X \in \mathbb{R}^{B \times L \times D}$, router weights $W_r \in \mathbb{R}^{D \times N}$, experts $\{E_i\}_{i=1}^N$, controlling factor $\alpha \in [0,1]$, score functions $S_t$ (TC) and $S_e$ (EC)
2: **Output:** Output tensor $Y \in \mathbb{R}^{B \times L \times D}$
3: $S \leftarrow XW_r$ {Compute routing logits via dot product}
4: $U \leftarrow (1-\alpha) \cdot S_t + \alpha \cdot S_e$ {Compute unified scores}
5: $U_{flat} \leftarrow \text{reshape}(U, [B, L \cdot N])$ {Flatten for global or sequence-level selection}
6: $V_{top}, I_{top} \leftarrow \text{TopK}(U_{flat}, k=n)$ {Select top-$n$ expert-token pairs}
7: $Y \leftarrow \text{SparseDispatcher}(X, \{E_i\}, V_{top}, I_{top})$ {Execute sparse MoE computation}
8: **return** $Y$

---

| Task | Original (TC) | EC | USMoE |
|---|---|---|---|
| ARC-C | 0.00 | 0.03 | 0.00 |
| ARC-E | 0.00 | 0.03 | 0.00 |
| BoolQ | 0.00 | 0.01 | 0.00 |
| OBQA | 0.00 | 0.02 | 0.00 |
| PIQA | 0.00 | 0.04 | 0.00 |
| WinoGrande | 0.00 | 0.05 | 0.00 |
| **Average** | **0.00** | **0.03** | **0.00** |

*Table 8.* Average token dropping rate of different routing mechanisms on Qwen3-30B-A3B-Instruct with $K = 8$. EC drops a non-trivial fraction of tokens, whereas TC and USMoE preserve all tokens in our evaluation.

Table 8 reports the average token dropping rate of Qwen3-30B-A3B-Instruct across six benchmarks. EC exhibits a non-trivial average token dropping rate of approximately $3\%$, with dropping rates ranging from $1\%$ to $5\%$ across tasks. In contrast, both the original TC router and USMoE achieve $0\%$ token dropping on all evaluated datasets. These results suggest that USMoE effectively avoids the information-loss issue of EC while maintaining sparse routing.

**Token dropping analysis.** We further analyze the token dropping behavior of USMoE under global Top-$n$ routing. Let $U \in \mathbb{R}^{L \times N}$ denote the unified routing score matrix for a sequence of $L$ tokens and $N$ experts, where $U_{\ell,e}$ is the score of assigning token $\ell$ to expert $e$. USMoE selects the global Top-$n$ entries of $U$, where we set $n = KL$ so that the total number of selected token–expert assignments matches standard Top-$K$ token-choice routing.

Among the $LN$ possible token–expert pairs, USMoE retains exactly $n = KL$ entries. Therefore, the marginal probability that an arbitrary token–expert pair is selected is

$$\rho = \frac{n}{LN} = \frac{K}{N}. \tag{29}$$

A token $\ell$ is dropped if none of its $N$ token–expert pairs is selected. Under a standard independence approximation over the selection events of the $N$ experts for a given token, the token dropping probability is

$$\Pr(\mathcal{D}_\ell) = (1-\rho)^N = \left(1 - \frac{K}{N}\right)^N. \tag{30}$$

Using the inequality $(1-x)^N \le \exp(-Nx)$ for $x \in [0,1]$, we obtain

$$\Pr(\mathcal{D}_\ell) \le \exp\left(-N \cdot \frac{K}{N}\right) = \exp(-K). \tag{31}$$

Thus, the probability that a token receives no expert assignment decreases exponentially with the routing budget $K$. For the common setting $K = 8$, used by many sparse MoE models, this gives

$$\Pr(\mathcal{D}_\ell) \le e^{-8} \approx 3.35 \times 10^{-4}. \tag{32}$$

This bound explains why USMoE empirically exhibits zero observed token dropping across all evaluated tasks in Table 8. Unlike EC, which can systematically ignore low-ranked tokens from the expert perspective, USMoE performs global token–expert selection under a fixed assignment budget, making token dropping exponentially unlikely as $K$ increases.

## A.5. Experimental details and additional results

### A.5.1. SUPERVISED FINE-TUNING

**Settings.** We demonstrate the effectiveness of USMoE on supervised fine-tuning using the Alpaca dataset (Taori et al., 2023), adopting the memory-efficient full-parameter training approach proposed by (Zhao et al., 2024), which outperforms common low-rank adaptation methods in terms of memory efficiency. To evaluate the robustness of our method, we follow the procedure in (Nielsen et al., 2025), applying a Text Attack strategy where words are randomly replaced with a generic token "AAA".

**Supervised Fine-Tuning.** We demonstrate the efficiency and robustness of USMoE in supervised fine-tuning across two advanced SMoE models: QwenMoE (7B)(Team, 2024) and OLMoE (7B)(Muennighoff et al., 2025). For training, we adopt Supervised Fine-Tuning (SFT) using GaLore(Zhao et al., 2024), which enables full-parameter optimization with improved memory efficiency compared to typical low-rank adaptation methods such as LoRA. The fine-tuning is conducted on the Alpaca dataset(Taori et al., 2023) for 2,000 steps, with evaluation performed every 10 steps. Furthermore, we assess our method on both clean (referred to as QwenMoE and OLMoE) and adversarially corrupted versions (denoted QwenMoE-Corrupt and OLMoE-Corrupt), where sequences of random "AAA" tokens are injected to simulate noise.

**Robustness.** Figure 6 presents the training results for both QwenMoE and OLMoE models. The results show that USMoE consistently outperforms the baselines across both QwenMoE and OLMoE models. Our method surpasses both Token Choice (TC) and Expert Choice (EC), particularly on QwenMoE, where it achieves significantly lower training and validation perplexity, indicating faster convergence and better generalization. Notably, the performance gap becomes even more pronounced under the corrupted setting, highlighting the superior robustness of USMoE compared to traditional SMoE approaches.

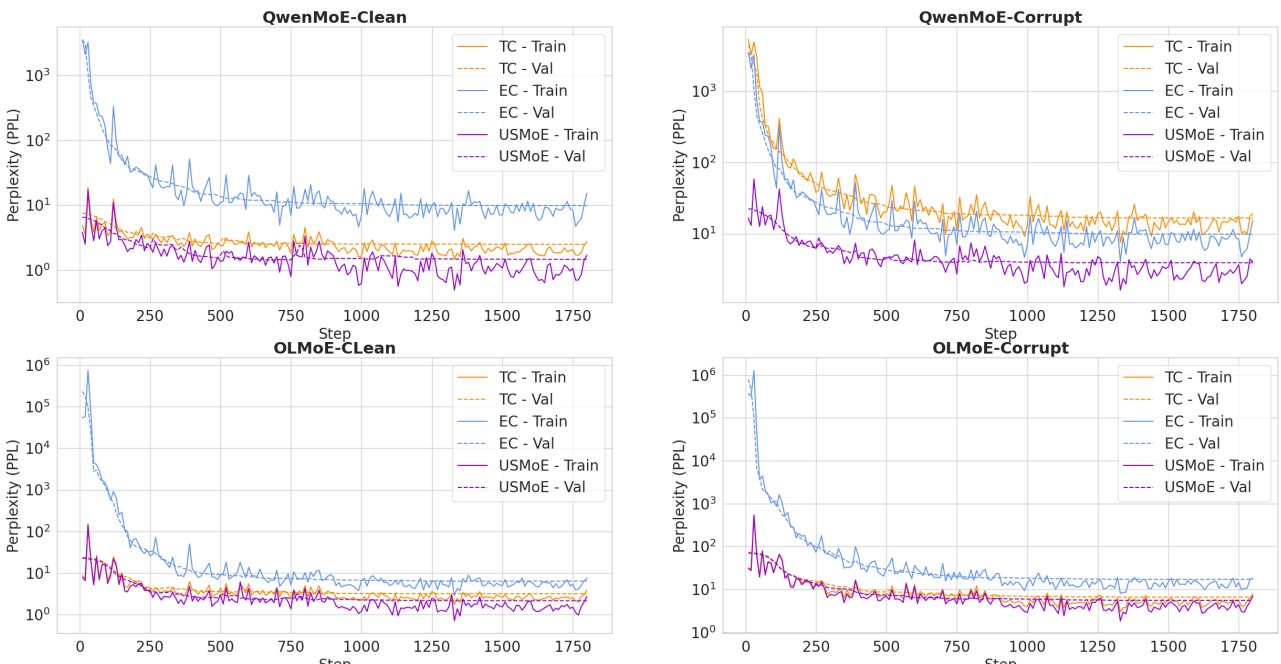

*Figure 6.* Illustration of comparing the performance of USMoE, Token Choice (TC), Expert Choice (EC) using QwenMoE and OLMoE for the Supervised Fine-Tuning task on Apaca dataset for 2K steps under both clean and corrupted settings. Training and validation perplexity over training steps are reported, and lower values are better.

A.5.2. WHICH MAPPING FUNCTION FOR EXPERT CHOICE?

**Revising Expert Choice.** The Expert Choice (EC) approach faces scalability challenges in autoregressive models due to information leakage (Wang et al., 2024; Raposo et al., 2024) introduced by the softmax operator. To address this, we replace softmax with a sigmoid function for Token Choice, which eliminates inter-token leakage and, as shown in Figure 7, leads to improved bits-per-character (BPC) performance and a reduced token dropping ratio.

The Expert Choice (EC) approach faces scalability challenges for large language models due to the issues of *information leakage* (Wang et al., 2024; Raposo et al., 2024) and *token dropping* (Puigcerver et al., 2024). We revisit this problem and identify that these issues stem from the use of the `softmax` operator, which introduces information leakage across tokens, and the `top-k` selection applied along the batch dimension.

To address this, we propose replacing the `softmax` mapping function with a `sigmoid` function to eliminate inter-token information leakage. Additionally, we mitigate batch-wise leakage by applying the `top-k` selection only along the sequence dimension.

Figure 7 compares the performance of EC using `softmax` versus `sigmoid`. The results show that the `sigmoid`-based EC not only achieves better bits-per-character (BPC) performance but also reduces the token dropping ratio. This improvement suggests that using the `sigmoid` mapping function has strong potential for scaling up the EC approach in large language models.

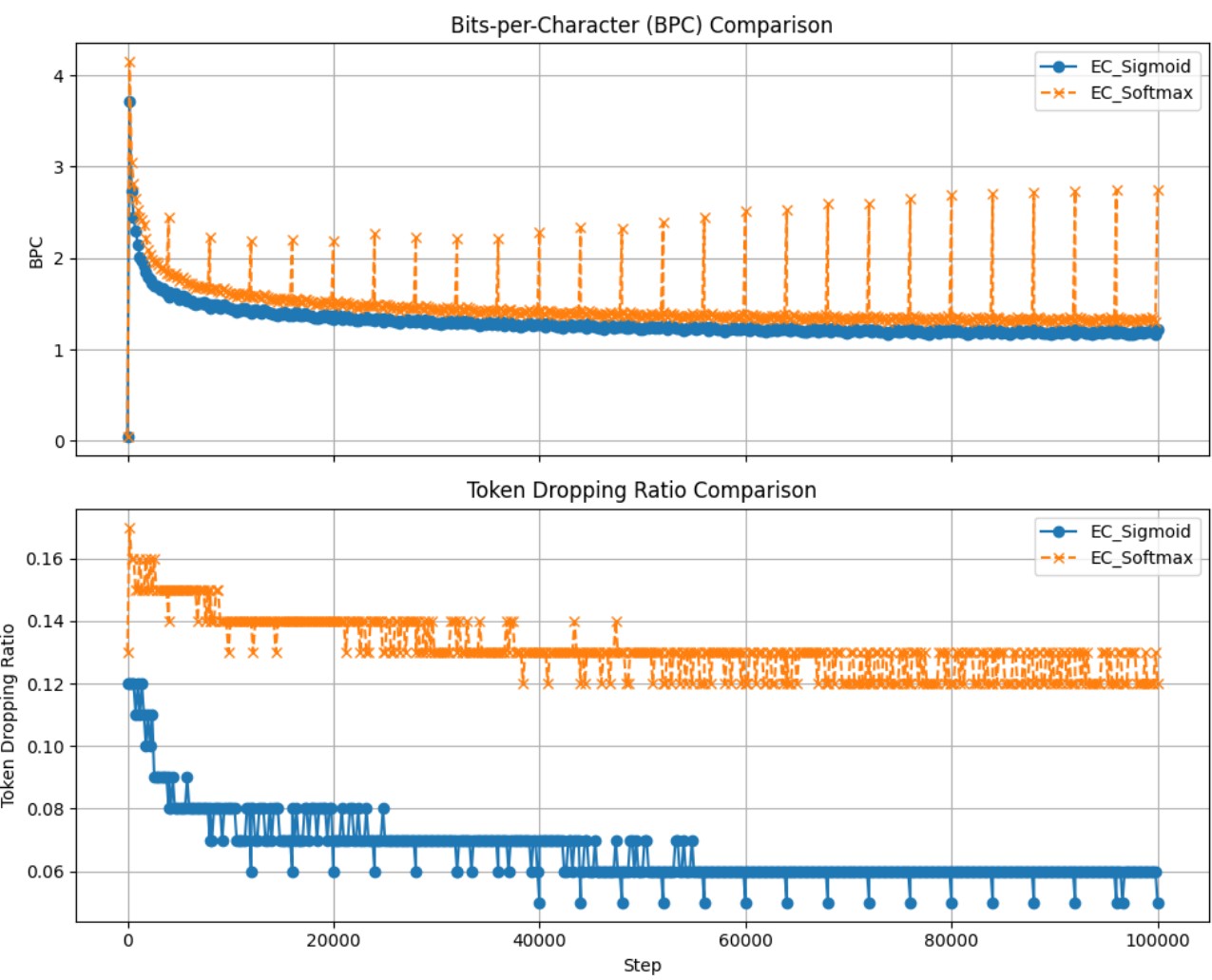

*Figure 7.* Performance comparison of the Expert Choice approach using two mapping functions: `sigmoid` and `softmax`, evaluated on training performance and token dropping ratio (lower is better).

| Setting | Data | Top-$K$ | TC (Original) | USMoE |
|---------|------|---------|---------------|-------|
| Corrupt | BoolQ | 1 | 0.418 | **0.455** |
| Corrupt | BoolQ | 2 | 0.483 | **0.493** |
| Corrupt | BoolQ | 4 | 0.489 | **0.693** |
| Corrupt | BoolQ | 6 | 0.490 | **0.784** |
| Corrupt | BoolQ | 8 | 0.600 | **0.808** |

*Table 9.* Scalability under varying Top-$K$ expert budgets on the corrupted BoolQ setting with Qwen3-30B-A3B-Instruct. USMoE consistently outperforms standard Token Choice (TC) routing across all compute budgets.

| Method | Context Length | Doc Depth | Retrieval Acc. |
|--------|----------------|-----------|----------------|
| Original (TC) | 16.5K | 50% | 50% |
| EC | 16.5K | 50% | 10% |
| USMoE | 16.5K | 50% | **100%** |

*Table 10.* Needle-in-a-Haystack retrieval accuracy on Qwen3-30B-A3B-Instruct. The context length is 16.5K tokens, and the needle is inserted at 50% document depth.

### A.5.3. SCALABILITY UNDER VARYING TOP-$K$ BUDGETS

Beyond evaluating USMoE across model scales, ranging from 20M to 30B parameters, we further study whether its advantage persists under constrained expert-computation budgets. Specifically, we vary the number of selected experts per token, $K \in \{1, 2, 4, 6, 8\}$, and compare USMoE against standard Token Choice (TC) routing on the corrupted BoolQ setting using Qwen3-30B-A3B-Instruct.

As shown in Table 9, USMoE consistently outperforms TC across all Top-$K$ budgets. The improvement is modest when the routing budget is extremely sparse, e.g., $K = 1$ or $K = 2$, but becomes substantially larger as the available expert budget increases. In particular, at $K = 4$, USMoE improves accuracy from $0.489$ to $0.693$, and the gap continues to widen at $K = 6$ and $K = 8$. This trend suggests that USMoE is better able to exploit additional expert capacity by allocating the global token–expert budget to more informative token–expert pairs, rather than enforcing an identical local Top-$K$ budget for every token.

These results demonstrate that the benefit of USMoE is not limited to a single routing configuration. Instead, USMoE remains robust across different compute budgets and scales favorably as more experts are made available. This supports our hypothesis that global token–expert allocation provides a more effective use of sparse expert computation than token-wise expert selection.

### A.5.4. LONG-CONTEXT RETRIEVAL BENEFIT

We evaluate USMoE on the Needle-in-a-Haystack task (Kamradt, 2023), which measures whether a model can retrieve a target fact inserted into a long distractor-heavy context. As shown in Table 10, using Qwen3-30B-A3B-Instruct with a 16.5K-token context and the needle placed at 50% document depth, USMoE achieves 100% retrieval accuracy. This substantially improves over both the original Token Choice (TC) router, which obtains 50%, and Expert Choice (EC), which obtains 10%.

These results indicate that USMoE improves long-context retrieval by more effectively allocating sparse expert computation to informative token–expert pairs. By combining token-wise and expert-wise routing signals, USMoE better preserves rare but task-critical evidence in distraction-heavy contexts.

### A.5.5. LOAD-BALANCING ANALYSIS

Following Token Choice routing, we include a standard load-balancing loss to mitigate expert imbalance. Table 11 reports results on Transformer-XL (20M) trained on Enwik8. USMoE obtains a lower load-balancing loss than TC, decreasing from 1.64 to 0.98, which indicates a more uniform distribution of tokens across experts. USMoE also improves language modeling performance, reducing BPC from 1.20 to 1.18. These results suggest that USMoE not only improves expert

| Model | Data | Metric | TC | USMoE |
|---|---|---|---|---|
| Transformer-XL (20M) | Enwik8 | Load-balancing Loss ↓ | 1.64 | **0.98** |
| Transformer-XL (20M) | Enwik8 | BPC ↓ | 1.20 | **1.18** |

*Table 11.* Load-balancing analysis on Transformer-XL (20M) using Enwik8. USMoE achieves lower balancing loss and better BPC than Token Choice (TC).

| Task | TC Acc. | USMoE Acc. | TC Thinking Tokens | USMoE Thinking Tokens |
|---|---|---|---|---|
| ARC-C | 0.584 | **0.600** | 586 | **565** |
| ARC-E | 0.819 | **0.822** | 353 | **346** |
| BoolQ | 0.866 | **0.870** | 508 | **484** |
| OBQA | 0.424 | **0.450** | 630 | **583** |
| PIQA | 0.812 | **0.842** | **638** | 664 |
| WinoGrande | 0.729 | **0.739** | 701 | **671** |
| Average | 0.706 | **0.721** | 569 | **552** |

*Table 12.* Performance of USMoE on Qwen3-30B-A3B-Thinking. USMoE improves average reasoning accuracy while reducing the average number of thinking tokens.

utilization but can also translate better load balancing into improved modeling quality.

### A.5.6. PERFORMANCE ON REASONING MODELS

We further evaluate USMoE on Qwen3-30B-A3B-Thinking to examine its effect on reasoning-oriented models. As shown in Table 12, USMoE improves average accuracy from $0.706$ to $0.721$ over the original Token Choice (TC) router. Meanwhile, it reduces the average thinking length from $569$ to $552$ tokens. These results suggest that USMoE improves reasoning accuracy while maintaining, and slightly improving, inference efficiency. The reduction in thinking tokens also indicates that USMoE may mitigate unnecessary information propagation during reasoning.

### A.5.7. COMPARISON WITH ADVANCED MoE ROUTING METHODS

We compare USMoE with ReMoE (Wang et al., 2025), an advanced MoE routing method that dynamically reduces the number of activated experts. As shown in Table 13, USMoE substantially outperforms ReMoE across all tasks. Under the standard Top-$K$=8 setting, USMoE improves the average accuracy over the original TC baseline from $0.7245$ to $0.732$. In contrast, ReMoE obtains an average accuracy of $0.373$ with an average activated Top-$K$ of approximately $1.5$. When USMoE is evaluated under the same effective Top-$K$ budget as ReMoE, it still achieves a higher average accuracy of $0.444$. These results indicate that USMoE provides a stronger accuracy–efficiency trade-off than ReMoE.

### A.5.8. TRAINING-FREE ADDITIONAL RESULTS

In this section, we compare methods both with and without prompts, including PromptEOL (Jiang et al., 2024b). Inspired by (Li & Zhou, 2025), we also compare our method with an approach that uses the similarity score between router and expert embeddings as the hidden representation, which we refer to as "Router Embedding" or simply "Router" Additionally, we evaluate against MoEE (Li & Zhou, 2025), which leverages both Router Embedding and the hidden representation of the SMoE model as embeddings.

For tasks evaluated in Figure 9, USMoE proves even more effective at enhancing the Token Choice approach, delivering notable gains of **14%**, **12%**, and **13%** for *OLMoE-1B-7B*, *Qwen1.5-MoE-A2.7B*, and *DeepSeekMoE-16B*, respectively, across MTEB tasks without additional training. Specifically, *Qwen1.5-MoE-A2.7B* achieves a remarkable improvement from 13.4% (Token Choice) to 40.0% (USMoE) in the Summarization task, representing a **198%** gain. This trend persists across *DeepSeekMoE-16B* and *OLMoE-1B-7B*, where USMoE consistently outperforms both the Token Choice and Expert Choice approaches. Overall, our approach outperforms the baselines in terms of performance while exhibiting lower variance across multiple tasks and different runs. Our method consistently demonstrates performance improvements across a range of MTEB tasks .

| Task | TC Top-$K$=8 | ReMoE | USMoE Top-$K$=8 | USMoE Top-$K$≈ReMoE | ReMoE Actual Top-$K$ |
|---|---|---|---|---|---|
| ARC-C | 0.631 | 0.241 | **0.646** | 0.265 | 1.4 |
| ARC-E | 0.838 | 0.258 | **0.846** | 0.352 | 1.3 |
| BoolQ | 0.886 | 0.444 | **0.894** | 0.587 | 1.6 |
| OBQA | 0.454 | 0.266 | **0.460** | 0.274 | 1.5 |
| PIQA | 0.805 | 0.515 | **0.810** | 0.574 | 1.3 |
| WinoGrande | 0.733 | 0.516 | **0.736** | 0.615 | 1.6 |
| Average | 0.725 | 0.373 | **0.732** | 0.444 | 1.5 |

*Table 13.* Comparison between USMoE and ReMoE (Wang et al., 2025) on Qwen3-30B-A3B-Instruct. USMoE outperforms ReMoE both at Top-$K$=8 and under a comparable effective Top-$K$ budget.

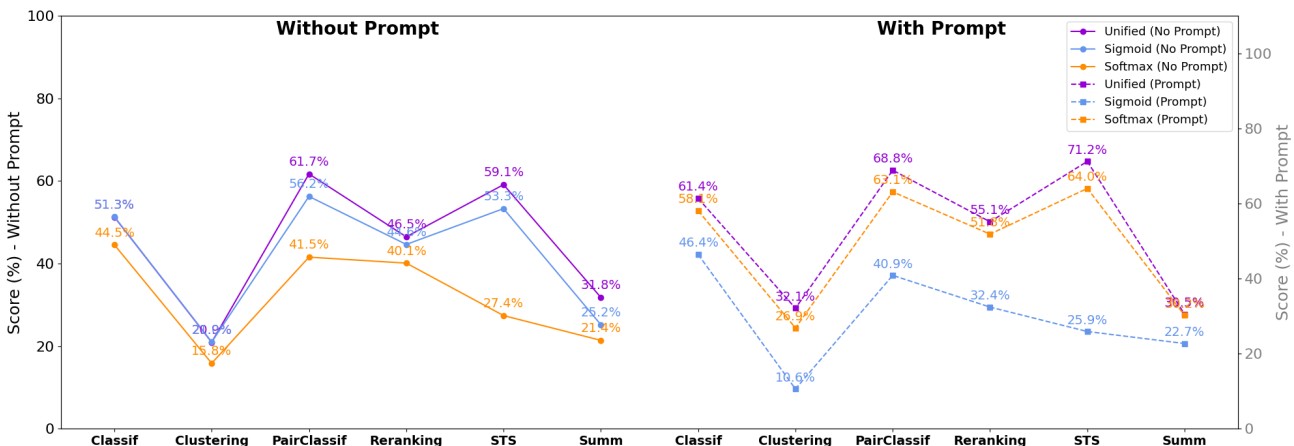

*Figure 8.* The performance of Unified Score (USMoE), Softmax (TC), and Sigmoid (EC) across the MTEB benchmark. Sigmoid outperforms Softmax on tasks without prompting, indicating stronger semantic representations.

Interestingly, Router Embedding is less affected by prompting on the Classification dataset, as shown in Figure 11a, while Token Choice, Expert Choice, and USMoE (ours) achieve significant performance improvements in the prompting setting. Figure 11a also demonstrates that our method is not only more effective but also more stable than the baselines, as indicated by a lower *standard deviation*. Additionally, Figure 11b illustrates the distribution of our method across MTEB tasks in both prompted and non-prompted scenarios. Overall, our approach outperforms the baselines in terms of performance while exhibiting lower variance across multiple tasks and different runs.

We provide a detailed evaluation of three state-of-the-art SMoE models: **OLMoE-1B-7B** (Table 15), **Qwen1.5-MoE-A2.7B** (Table 16), and **DeepSeekMoE-16B** (Table 17). Our results demonstrate the effectiveness of our method across various models and prompts, comparing its performance against baseline approaches such as Token Choice (TC) and Expert Choice (EC).

### A.5.9. TRAINING FROM SCRATCH

**Large models training.** USMoE not only delivers strong performance in base model training but also remains highly competitive at a large scale. Table 18 presents perplexity (PPL) results on the `WikiText-103` and `One Billion Words` datasets using a large Transformer-XL model with *15 SMoE layers*, *16 experts*, and *420M parameters*. The performance gap between USMoE and the baselines becomes even more pronounced at this scale, highlighting its strong scalability with increasing model complexity. Regardless of backbone size or the number of activated experts, USMoE consistently outperforms all baselines, demonstrating its effectiveness in scaling up large language models.

**Training from scratch.** To assess the effectiveness of our method, we compare USMoE with the Token Choice approaches, including SMoE (Jiang et al., 2024a), SMoE-Dropout (abbreviated as "SMoE-DR"), XMoE (Chi et al., 2022), and StableMoE (Dai et al., 2022), as well as the Expert Choice approach (Zhou et al., 2022) for pre-training and fine-tuning tasks. We follow the approach of (Chen et al., 2023) and use a base Transformer-XL (Dai et al., 2019) with four decoder

| Model | Task | Router | TC | EC | MoEE | USMoE |
|---|---|---|---|---|---|---|
| | Classification | 41.2 | 43.4 | 44.9 | 41.8 | **51.3** |
| | Clustering | 13.7 | 14.7 | 12.0 | 14.5 | **21.0** |
| | PairClassification | 45.3 | 39.1 | 35.5 | 45.7 | **61.7** |
| OLMoE-1B-7B | Reranking | 37.5 | 37.4 | 35.3 | 39.5 | **46.5** |
| | STS | 39.9 | 24.1 | 18.2 | 39.9 | **59.6** |
| | Summarization | 28.4 | 20.9 | 21.1 | 29.8 | **31.8** |
| | **Average** | 34.3 | 29.9 | 27.8 | 35.2 | **45.3** |
| | Classification | 43.8 | 50.3 | 25.5 | 47.7 | **52.2** |
| | Clustering | 13.6 | 27.4 | 23.2 | 25.2 | **29.9** |
| | PairClassification | 45.9 | 46.9 | 43.4 | 51.5 | **56.2** |
| Qwen1.5-MoE-A2.7B | Reranking | 39.6 | 45.3 | 41.6 | 48.5 | **53.1** |
| | STS | 38.8 | 38.0 | 35.6 | 51.8 | **60.7** |
| | Summarization | 28.3 | 13.4 | 15.1 | 31.2 | **40.0** |
| | **Average** | 35.0 | 36.9 | 30.7 | 42.6 | **48.7** |
| | Classification | 43.4 | 46.6 | 44.7 | 44.4 | **49.4** |
| | Clustering | 13.4 | 18.1 | 13.5 | 17.8 | **21.0** |
| | PairClassification | 45.5 | 40.9 | 37.1 | 46.1 | **53.5** |
| DeepSeekMoE-16B | Reranking | 38.5 | 38.9 | 35.1 | 42.2 | **45.7** |
| | STS | 37.7 | 26.3 | 23.3 | 40.2 | **51.2** |
| | Summarization | 24.9 | 22.0 | 18.5 | 24.4 | **29.9** |
| | **Average** | 33.9 | 32.1 | 28.7 | 35.9 | **41.8** |

*Table 14.* Performance comparison of USMoE, Token Choice (TC), Expert Choice (EC), and MoEE across MTEB Tasks **without prompts** and models. The best result for each row is highlighted in **bold**.

layers. We train both base and large-scale versions of Transformer-XL on four datasets (Enwik8, Text8, Wikitext-103, and One Billion Words) for 100k iterations, following the implementation in (Chen et al., 2023). Then we fine-tune the pre-trained weights for text classification tasks, including SST-2 (Socher et al., 2013), SST-5 (Socher et al., 2013), IMDB (Maas et al., 2011), and BANKING77 (Casanueva et al., 2020). More implementation details and additional results are provided in the Appendix A.

**Fine-tuning.** We report the results of the fine-tuning experiment on the SST-2, SST-5, IMDB, and BANKING77 datasets in Table 19, using Transformer-XL pre-trained on enwik8. Overall, USMoE consistently achieves higher accuracy compared to other baselines across all datasets. The results demonstrate that our method is not only effective for pre-training tasks but also performs effectively on existing pre-trained models.

### A.5.10. ROBUSTNESS FOR VISION TASKS

**Robustness.** Table 20 presents a comparative analysis of ViT-MoE variants, USMoE, Token Choice (TC), Expert Choice (EC), and SoftMoE, under three common adversarial attacks: Projected Gradient Descent(PGD)(Chang et al., 2018), Fast Gradient Sign Method(FGSM) (Goodfellow et al., 2015), and Simultaneous Perturbation Stochastic Approximation (SPSA) (Spall, 1997), across five standard image classification datasets. USMoE consistently achieves strong performance, outperforming or matching the best baselines in most scenarios. Notably, it achieves the highest average robustness across all three attack types, particularly excelling under PGD (46.2%), FGSM (43.3%), and SPSA (63.9%). While Token Choice (TC) performs poorly across all three attack types, USMoE consistently delivers stronger and more reliable performance on average. These results highlight USMoE's robustness and generalization capability under adversarial conditions, making it a more stable and effective choice for secure image classification tasks.

### A.6. In-depth Analysis

We visualize the router behavior of USMoE in Figure 15 and contrast it with the router behaviors of the Token Choice approach (Figure 13) and the Expert Choice approach (Figure 14). Notably, the router in the **OLMoE-1B-7B** model

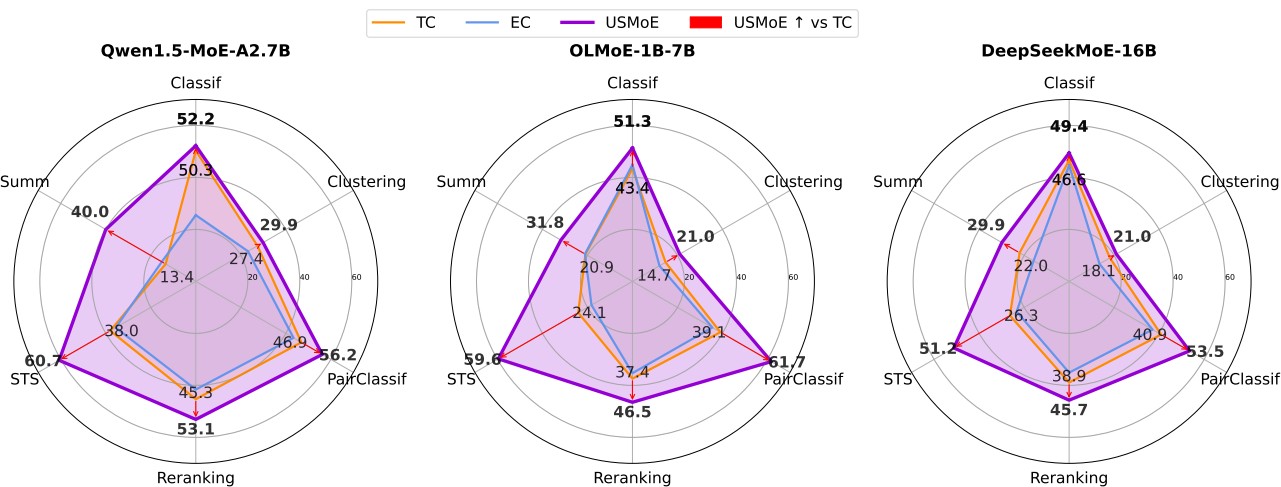

*Figure 9.* Performance comparison of USMoE, Token Choice (TC), Expert Choice (EC), and MoEE across MTEB Tasks and advance SMoE models. The best result for each row is highlighted in **bold**. Best viewed in color.

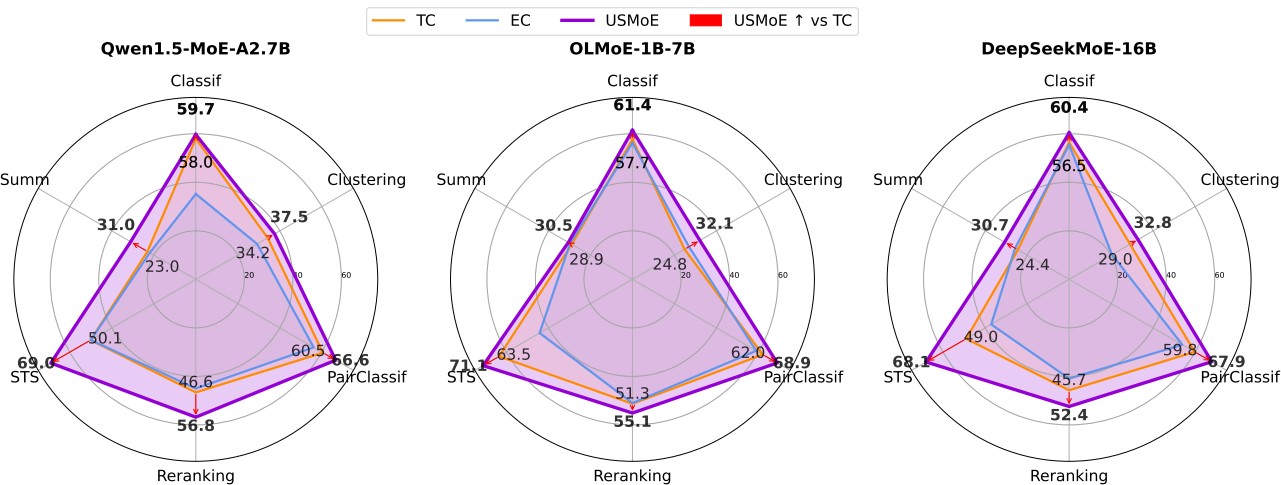

*Figure 10.* Performance comparison of USMoE, Token Choice (TC), Expert Choice (EC), and MoEE across across MTEB Tasks **with PromptEOL (Jiang et al., 2024b)**. The best result for each row is highlighted in **bold**.

exhibits a strong preference for specific experts. For instance, in the *Emotion Classification* task, Experts 8, 30, and 58 are consistently prioritized in both the Token Choice and Expert Choice approaches. This bias limits the model's adaptability and effectiveness for downstream tasks. USMoE tackles this challenge by introducing the *Unified Mechanism*, which promotes more balanced and diverse expert selections, as illustrated in Figure 15. This enhancement enables USMoE to outperform the baselines on the *Emotion Classification* task.

We track the number of unique experts utilized by the **OLMoE-1B-7B** model for each sequence in the *Emotion Classification* task as Figure 16a. Our analysis reveals that the Expert Choice approach employs 11 out of 16 experts, indicating a lower level of specialization among experts. In contrast, both USMoE and the Token Choice approach use an average of 0.9 to 1 expert per sequence, demonstrating superior expert specialization. Furthermore, we analyze the token dropping behavior of the Expert Choice approach and observe a significant increase in dropping rates when scaling to larger datasets or models, such as pre-training the Transformer-XL Large model on the *One Billion Word* dataset, as shown in Figure 16b. This increase in dropping rates may negatively impact model performance. In contrast, our method maintains a consistently low dropping rate (¡0.1), demonstrating its superiority over the Expert Choice approach for scalability. Additionally, our method proves more robust than the Token Choice approach, as it effectively drops irrelevant tokens without compromising performance.

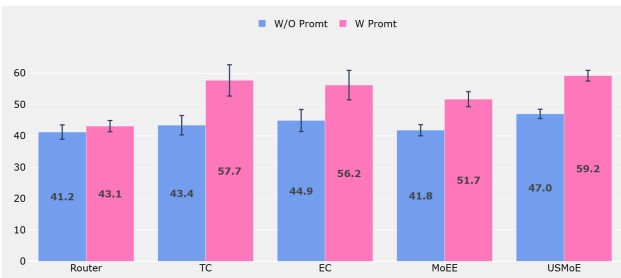

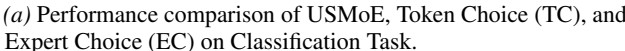

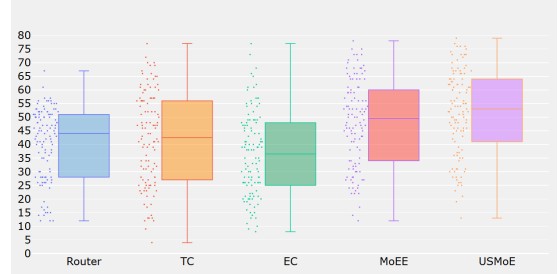

*(a)* Performance comparison of USMoE, Token Choice (TC), and Expert Choice (EC) on Classification Task.

*(b)* Results distribution of USMoE, Token Choice (TC), Expert Choice (EC), and MoEE across MTEB Tasks

*Figure 11.* Illustration of comparing the performance of USMoE, Token Choice (TC), Expert Choice (EC), and MoEE across MTEB tasks and three SMoE models. Each benchmark is run 10 times, reporting both the mean and standard deviation to highlight the performance and stability of our method compared to the baselines.

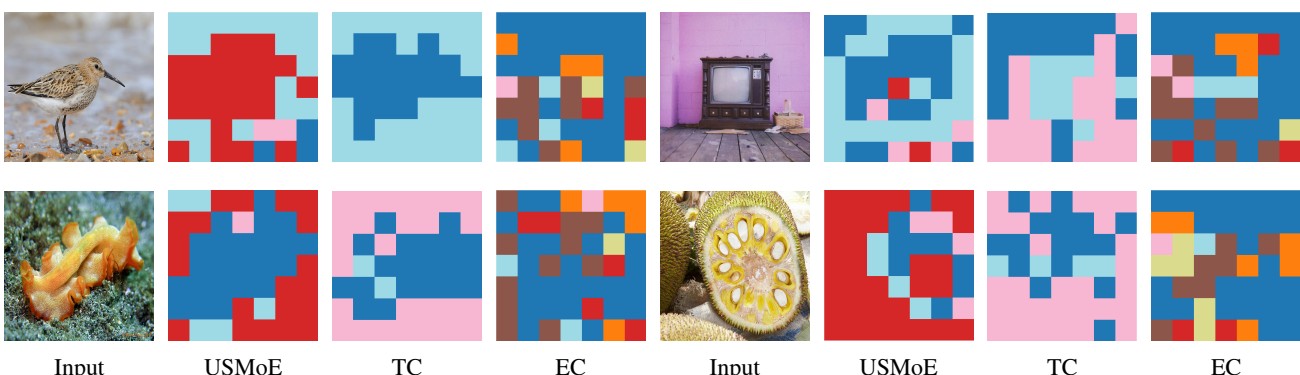

| Input | USMoE | TC | EC | Input | USMoE | TC | EC |

*Figure 12.* We compare token routing performance in vision tasks using 7×7 images, where each token is color-coded based on its assigned expert under setting $c = t$, where $c$ is a sparsity constraint and $t$ is number of image patches. **Left:** When the object is easy to distinguish, **Expert Choice** (EC) fails to assign different experts appropriately. **Token Choice** (TC) performs better but still does not align perfectly with the actual object, while USMoE correctly separates the object. **Right:** In more challenging images, both Expert Choice (EC) and Token Choice (TC) fail to distinguish between object and background. In contrast, USMoE successfully differentiates the object from the background, demonstrating greater efficiency in vision tasks compared to EC and TC, as further shown in Section 4.

.

## A.7. Implementation Details

For the **Without Training** experiments, we implement our method based on the publicly available MoEE implementation (Li & Zhou, 2025)[1]. Due to resource constraints, we validate our method and the baselines using 4-bit quantization with a batch size of 128. For the *OLMoE-1B-7B* model, we conduct experiments on a single H100 GPU, while for the *Qwen1.5-MoE-A2.7B* and *DeepSeekMoE-16B* models, we utilize two H100 GPUs.

The base Transformer-XL variant (Chen et al., 2023) comprises four Transformer decoder layers, each with an input dimension of 256. Each layer includes a self-attention mechanism with eight attention heads, followed by a Feed-forward Neural Network (FFN) that has an inner dimension of 512. The dropout ratio is set at 0.1. We divide the FFN into 16 experts, each with the same dimensions. For the larger variants, we scale the model up to twelve layers.

Our experiments are based on the publicly available SMoE-Dropout implementation (Chen et al., 2023)[2]. The pre-training experiments were conducted using a single H100 GPU, while the fine-tuning experiments were performed on a single A100 GPU. It is important to note that parallel training on multiple GPUs may produce different results.

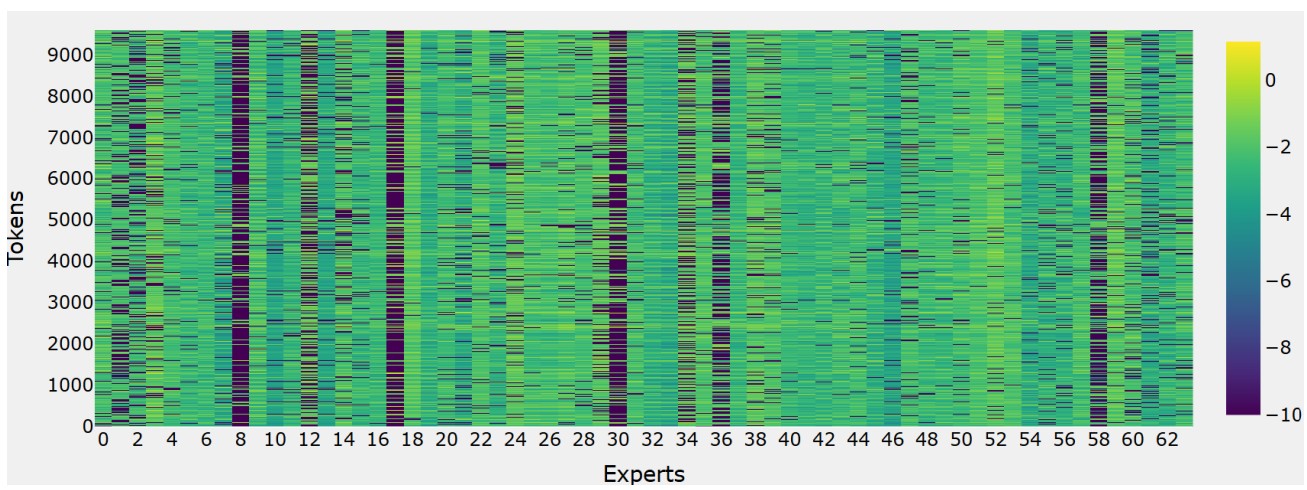

*Figure 13.* **Token Choice Router** visualization for the **OLMoE-1B-7B** model on the *Emotion Classification* task. The scores of selected experts are replaced with *-10.0* (lower than the minimum score) to enhance visualization. Best viewed in color.

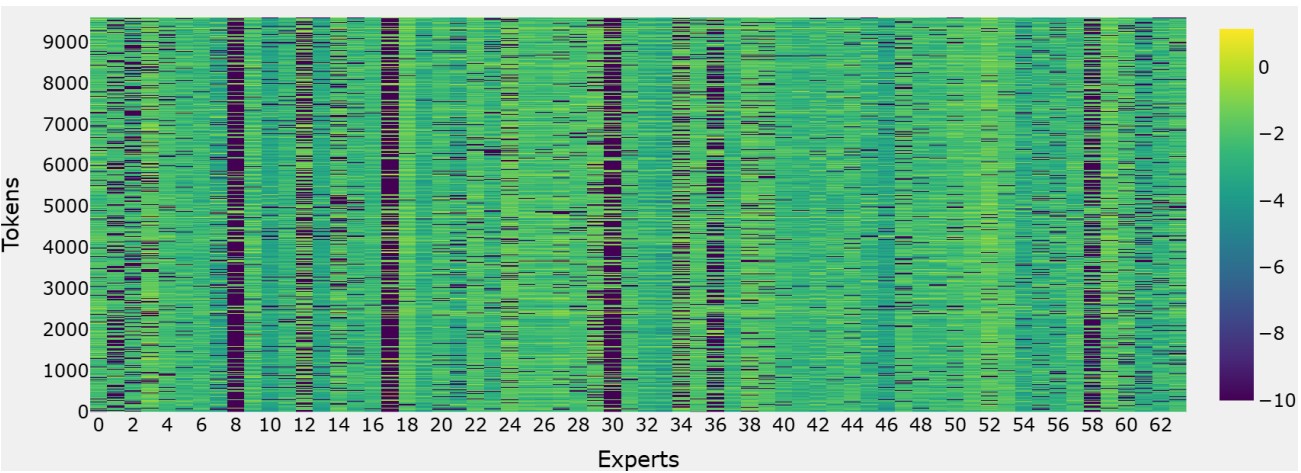

*Figure 14.* **Expert Choice Router** visualization for the **OLMoE-1B-7B** model on the *Emotion Classification* task. The scores of selected experts are replaced with *-10.0* (lower than the minimum score) to enhance visualization. Best viewed in color.

### A.7.1. PRE-TRAINING EXPERIMENTS

We provide the USMoE implementation details for pre-training our Transformer-XL base and large on `enwik8`, `text8`, `WikiText-103`, and `One Billion Word` in Table 22.

### A.7.2. FINE-TUNING EXPERIMENTS

To perform the fine-tuning experiments, we utilize the same model architecture as in the pre-training phase. Table 23 presents the implementation details for the fine-tuning experiments conducted across four different datasets.

---

[1]https://github.com/tianyi-lab/MoE-Embedding
[2]https://github.com/VITA-Group/Random-MoE-as-Dropout

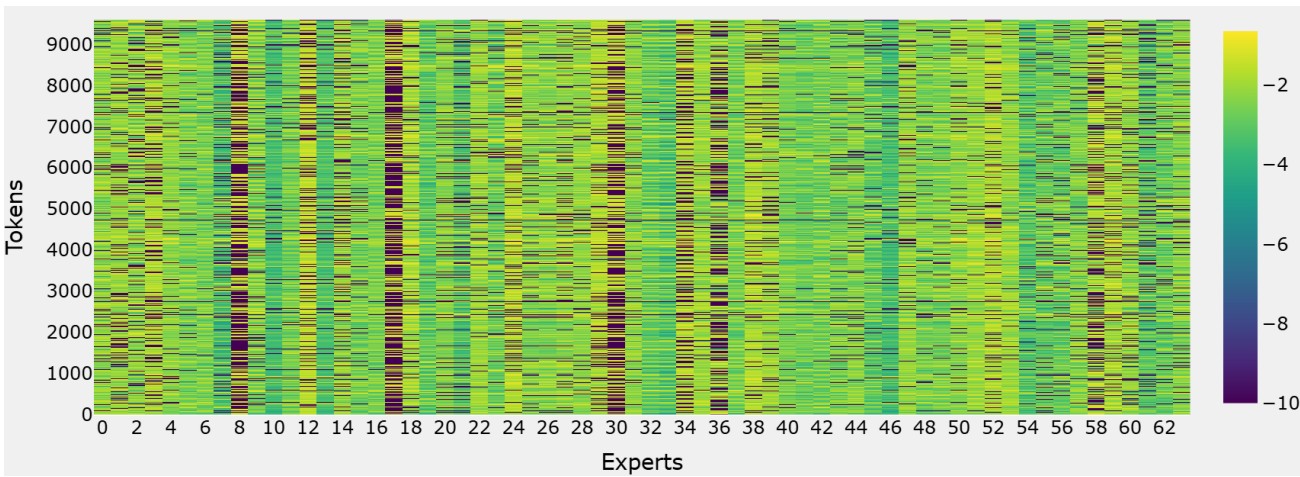

*Figure 15.* **USMoE Router** visualization for the **OLMoE-1B-7B** model on the *Emotion Classification* task. The scores of selected experts are replaced with *-10.0* (lower than the minimum score) to enhance visualization. Best viewed in color.

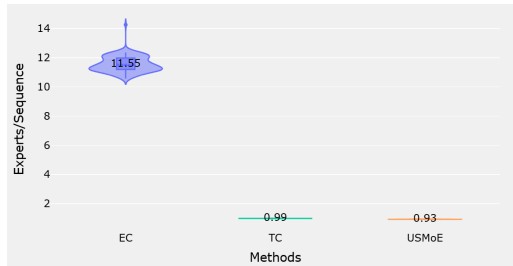

*(a)* Number Experts per Sequence of USMoE, TC, and EC on *Emotion* dataset.

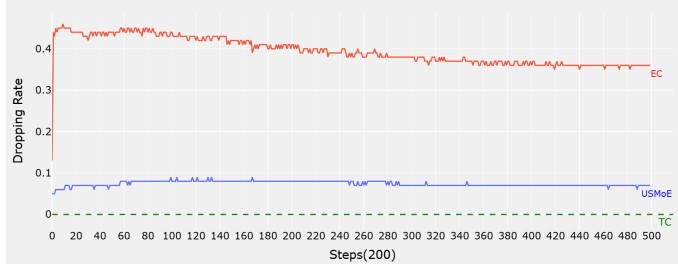

*(b)* Token Dropping of USMoE, Token Choice (TC), Expert Choice (EC) for Pre-training on *One Billion Word* dataset.

*Figure 16.* Comparison of the number of experts per sequence for USMoE, Token Choice (TC), and Expert Choice (EC) on the *Emotion* dataset using the **OLMoE-1B-7B** model, along with a comparison of token dropping rates for USMoE, TC, and EC during pre-training on the *One Billion Word* dataset.

| Category | Model | Dataset | Setting | Router | TC | EC | MoEE | USMoE |
|---|---|---|---|---|---|---|---|---|
| Classification | OLMoE | Emotion | None | 24.1 | 24.5 | 26.3 | 25.1 | **35.8** |
| | | | Prompt | 27.6 | 49.9 | 49.0 | 44.5 | **54.8** |
| | | Toxic | None | 51.9 | 58.9 | 59.9 | 51.9 | **62.8** |
| | | | Prompt | 52.3 | 65.2 | 61.3 | 53.4 | **67.6** |
| | | Tweet | None | 47.7 | 46.8 | 48.3 | 48.4 | **55.2** |
| | | | Prompt | 49.5 | 58.0 | 58.4 | 57.2 | **61.7** |
| Clustering | OLMoE | Medrxiv | None | 15.0 | 17.6 | 14.8 | 17.4 | **20.6** |
| | | | Prompt | 15.8 | 23.9 | 27.7 | 22.0 | **28.2** |
| | | 20Groups | None | 12.4 | 11.8 | 9.2 | 11.5 | **21.3** |
| | | | Prompt | 16.7 | 25.7 | 26.2 | 24.4 | **36.0** |
| Pair Classification | OLMoE | SemEval | None | 43.6 | 35.8 | 31.3 | 43.6 | **50.2** |
| | | | Prompt | 45.7 | 46.7 | 40.9 | 53.8 | **55.4** |
| | | URLCorpus | None | 47.0 | 42.4 | 39.7 | 47.8 | **73.2** |
| | | | Prompt | 61.4 | 77.4 | 76.9 | 78.2 | **82.3** |
| Reranking | OLMoE | Ask | None | 41.3 | 41.0 | 39.0 | 41.4 | **47.1** |
| | | | Prompt | 43.4 | 51.9 | 49.9 | 50.2 | **51.6** |
| | | SciDocs | None | 45.5 | 46.3 | 46.9 | 50.8 | **59.1** |
| | | | Prompt | 53.6 | 69.6 | 73.1 | 75.1 | **77.2** |
| | | StackOver | None | 25.8 | 24.8 | 20.1 | 26.4 | **33.2** |
| | | | Prompt | 28.1 | 32.5 | 30.0 | 34.3 | **36.6** |
| STS | OLMoE | Biosses | None | 39.3 | 13.6 | 7.7 | 29.7 | **64.0** |
| | | | Prompt | 51.2 | 61.8 | 67.6 | 70.2 | **75.2** |
| | | SickR | None | 50.3 | 46.3 | 26.4 | 53.0 | **58.9** |
| | | | Prompt | 51.9 | 65.7 | 37.6 | 66.1 | **66.5** |
| | | STS12 | None | 40.1 | 8.6 | 11.1 | 37.8 | **58.2** |
| | | | Prompt | 51.3 | 53.8 | 37.5 | 63.6 | **66.4** |
| | | STS13 | None | 40.5 | 21.1 | 18.2 | 43.4 | **61.5** |
| | | | Prompt | 52.5 | 66.5 | 40.4 | 72.7 | **76.4** |
| | | STS14 | None | 29.5 | 13.4 | 13.3 | 31.7 | **52.9** |
| | | | Prompt | 41.1 | 56.8 | 33.9 | 64.2 | **68.2** |
| | | STS15 | None | 30.8 | 27.8 | 22.5 | 33.3 | **63.2** |
| | | | Prompt | 46.4 | 69.3 | 38.4 | 66.4 | **72.5** |
| | | STS16 | None | 46.5 | 38.9 | 28.9 | 45.8 | **60.7** |
| | | | Prompt | 52.4 | 70.1 | 49.4 | 68.3 | **71.8** |
| | | STSBen | None | 42.2 | 23.4 | 17.5 | 44.5 | **57.4** |
| | | | Prompt | 48.6 | 63.6 | 48.9 | 70.7 | **72.1** |
| Summarization | OLMoE | Medrxiv | None | 28.4 | 20.9 | 21.1 | 29.8 | **31.8** |
| | | | Prompt | 25.6 | 28.9 | 29.7 | 30.4 | **30.5** |

*Table 15.* Performance comparison of USMoE, Token Choice (TC), Expert Choice (EC), and MoEE across MTEB tasks using the OLMoE model. The best result for each row is highlighted in **bold**.

| Category | Model | Dataset | Setting | Router | TC | EC | MoEE | USMoE |
|---|---|---|---|---|---|---|---|---|
| Classification | Qwen1.5-MoE-A2.7B | Emotion | None | 27.2 | 33.9 | 30.1 | 34.3 | **37.1** |
| | | | Prompt | 37.0 | 48.5 | 47.4 | 47.2 | **51.3** |
| | | Toxic | None | 53.0 | 61.1 | 21.3 | 52.9 | **61.4** |
| | | | Prompt | 53.4 | 64.5 | 19.8 | 54.1 | **65.5** |
| | | Tweet | None | 51.1 | 55.9 | 25.1 | 55.9 | **58.2** |
| | | | Prompt | 56.1 | 61.1 | 38.6 | 60.7 | **62.3** |
| Clustering | Qwen1.5-MoE-A2.7B | Medrxiv | None | 15.3 | 23.3 | 21.3 | 23.0 | **25.3** |
| | | | Prompt | 14.2 | 24.6 | 19.8 | 21.8 | **27.9** |
| | | 20Groups | None | 12.0 | 31.5 | 25.1 | 27.4 | **34.4** |
| | | | Prompt | 14.4 | 43.8 | 38.6 | 38.4 | **47.0** |
| Pair Classification | Qwen1.5-MoE-A2.7B | SemEval | None | 42.0 | 38.8 | 34.7 | 42.5 | **47.5** |
| | | | Prompt | 47.0 | 52.4 | 46.3 | 52.4 | **57.7** |
| | | URLCorpus | None | 49.8 | 54.9 | 52.1 | 60.6 | **64.8** |
| | | | Prompt | 56.7 | 68.7 | 65.8 | 68.2 | **75.5** |
| Reranking | Qwen1.5-MoE-A2.7B | Ask | None | 43.1 | 45.8 | 43.4 | 47.3 | **49.1** |
| | | | Prompt | 43.3 | 48.3 | 49.1 | 49.5 | **52.8** |
| | | SciDocs | None | 49.6 | 60.6 | 55.3 | 67.0 | **71.6** |
| | | | Prompt | 50.9 | 60.1 | 55.8 | 68.7 | **73.0** |
| | | StackOver | None | 26.2 | 29.5 | 26.2 | 31.1 | **38.5** |
| | | | Prompt | 28.8 | 31.3 | 30.2 | 35.2 | **44.5** |
| STS | Qwen1.5-MoE-A2.7B | Biosses | None | 33.8 | 32.5 | 34.7 | 49.6 | **66.6** |
| | | | Prompt | 55.1 | 55.8 | 48.5 | 68.4 | **74.5** |
| | | SickR | None | 51.0 | 55.5 | 40.4 | 61.0 | **63.5** |
| | | | Prompt | 50.2 | 59.7 | 51.1 | 64.3 | **69.1** |
| | | STS12 | None | 40.2 | 16.9 | 18.6 | 46.3 | **52.3** |
| | | | Prompt | 49.3 | 25.0 | 31.8 | 59.2 | **62.5** |
| | | STS13 | None | 38.1 | 42.9 | 44.2 | 56.7 | **67.3** |
| | | | Prompt | 53.3 | 57.5 | 54.6 | 73.4 | **75.6** |
| | | STS14 | None | 28.1 | 26.5 | 25.6 | 45.4 | **53.7** |
| | | | Prompt | 40.4 | 38.8 | 40.7 | 60.0 | **64.8** |
| | | STS15 | None | 34.8 | 40.5 | 38.4 | 46.1 | **56.1** |
| | | | Prompt | 40.7 | 52.3 | 54.2 | 58.8 | **66.8** |
| | | STS16 | None | 47.6 | 51.0 | 48.1 | 58.1 | **64.4** |
| | | | Prompt | 51.6 | 64.2 | 65.1 | 65.7 | **68.7** |
| | | STSBen | None | 37.0 | 37.7 | 34.7 | 50.9 | **61.6** |
| | | | Prompt | 45.6 | 47.8 | 54.5 | 64.5 | **70.0** |
| Summarization | Qwen1.5-MoE-A2.7B | Medrxiv | None | 28.3 | 13.4 | 15.1 | 31.2 | **40.0** |
| | | | Prompt | 27.0 | 23.0 | 21.9 | 27.3 | **31.0** |

*Table 16.* Performance comparison of USMoE, Token Choice (TC), Expert Choice (EC), and MoEE across MTEB Tasks with *Qwen1.5-MoE-A2.7B* models. The best result for each row is highlighted in **bold**.

| Category | Model | Dataset | Setting | Router | TC | EC | MoEE | USMoE |
|---|---|---|---|---|---|---|---|---|
| Classification | DeepSeekMoE-16B | Emotion | None | 26.1 | 27.4 | 26.5 | 27.6 | **31.3** |
| | | | Prompt | 37.9 | 48.3 | 46.1 | 46.4 | **52.6** |
| | | Toxic | None | 53.3 | 60.4 | 58.1 | 53.1 | **61.7** |
| | | | Prompt | 53.1 | 62.4 | 62.5 | 53.6 | **67.5** |
| | | Tweet | None | 51.0 | 51.9 | 49.5 | 52.6 | **55.2** |
| | | | Prompt | 54.9 | 58.4 | 57.5 | 58.9 | **61.0** |
| Clustering | DeepSeekMoE-16B | Medrxiv | None | 15.1 | 23.0 | 17.3 | 22.0 | **25.7** |
| | | | Prompt | 17.0 | 25.7 | 20.9 | 24.0 | **27.9** |
| | | 20Groups | None | 11.7 | 13.2 | 9.7 | 13.7 | **16.2** |
| | | | Prompt | 18.6 | 32.3 | 19.8 | 33.0 | **37.6** |
| Pair Classification | DeepSeekMoE-16B | SemEval | None | 44.6 | 40.2 | 32.6 | 43.5 | **44.6** |
| | | | Prompt | 48.4 | 47.2 | 46.6 | 51.3 | **55.7** |
| | | URLCorpus | None | 46.4 | 41.7 | 41.6 | 48.6 | **62.4** |
| | | | Prompt | 66.5 | 72.4 | 61.1 | 75.4 | **80.0** |
| Reranking | DeepSeekMoE-16B | Ask | None | 41.7 | 41.1 | 40.1 | 42.3 | **44.9** |
| | | | Prompt | 43.5 | 43.8 | 44.7 | 46.9 | **50.6** |
| | | SciDocs | None | 48.2 | 50.6 | 44.7 | 57.1 | **61.9** |
| | | | Prompt | 58.3 | 65.6 | 55.3 | 72.6 | **72.9** |
| | | StackOver | None | 25.7 | 24.9 | 20.4 | 27.3 | **30.2** |
| | | | Prompt | 29.7 | 27.6 | 22.6 | 32.3 | **33.6** |
| STS | DeepSeekMoE-16B | Biosses | None | 29.5 | 31.7 | 27.7 | 26.8 | **56.6** |
| | | | Prompt | 47.0 | 40.1 | 41.5 | 57.6 | **67.2** |
| | | SickR | None | 50.4 | 47.4 | 29.4 | 53.1 | **60.1** |
| | | | Prompt | 56.0 | 61.9 | 38.7 | 65.8 | **68.0** |
| | | STS12 | None | 44.0 | 4.3 | 13.9 | 45.0 | **48.6** |
| | | | Prompt | 57.8 | 31.0 | 28.4 | 64.0 | **65.9** |
| | | STS13 | None | 36.0 | 28.4 | 27.5 | 41.1 | **50.7** |
| | | | Prompt | 55.3 | 56.0 | 41.2 | 70.9 | **76.1** |
| | | STS14 | None | 25.4 | 12.0 | 13.0 | 28.2 | **41.2** |
| | | | Prompt | 44.9 | 41.0 | 31.1 | 58.6 | **65.1** |
| | | STS15 | None | 34.8 | 33.9 | 25.6 | 38.7 | **45.0** |
| | | | Prompt | 49.7 | 46.5 | 33.0 | 58.5 | **64.1** |
| | | STS16 | None | 44.9 | 34.4 | 33.1 | 46.9 | **55.2** |
| | | | Prompt | 56.7 | 58.0 | 44.2 | 64.5 | **67.4** |
| | | STSBen | None | 36.6 | 18.3 | 15.8 | 42.1 | **51.9** |
| | | | Prompt | 54.9 | 57.7 | 39.0 | 67.8 | **71.0** |
| Summarization | DeepSeekMoE-16B | Medrxiv | None | 24.9 | 22.0 | 18.5 | 24.4 | **29.9** |
| | | | Prompt | 29.1 | 24.4 | 25.7 | 29.2 | **30.7** |

*Table 17.* Performance comparison of USMoE, Token Choice (TC), Expert Choice (EC), and MoEE across MTEB Tasks with *DeepSeekMoE-16B* models. The best result for each row is highlighted in **bold**.

| Transformer-XL(420M) | WikiText-103 | | | lm1b | | |
|---|---|---|---|---|---|---|
| Topk | TC | EC | USMoE | TC | EC | USMoE |
| 1 | 31.70 | 35.52 | **25.48** | 58.65 | 65.43 | **56.90** |
| 2 | 22.42 | 23.30 | **22.06** | 44.56 | 43.39 | **40.53** |
| 4 | 23.57 | 23.60 | **22.65** | 45.52 | 43.70 | **40.90** |
| 8 | 24.20 | 24.37 | **22.88** | 46.36 | 44.22 | **43.24** |

*Table 18.* Large Scale performance comparison of USMoE, Token Choice (TC), and Expert Choice (EC) across multiple datasets, with perplexity on the WikiText-103 and One Billion Word test sets. Lower values are better, with the best results highlighted in **bold**.

| Transformer-XL(20M) | | FLOPs(x10$^{10}$) | SST-2 | SST-5 | IMDB | BANKING77 |
|---|---|---|---|---|---|---|
| USMoE (Topk=2) | | 7.7620 | 81.5 | **40.1** | **88.5** | **87.8** |
| (Topk=1.5) | | **6.6753** | **83.8** | 39.6 | 88.3 | 83.0 |
| TC (Topk=2) | SMoE | 7.7620 | 77.1 | 35.1 | 84.4 | 69.2 |
| | SMoE-DR | | 78.6 | 34.4 | 83.5 | 66.7 |
| | XMoE | | 76.7 | 35.3 | 83.3 | 67.4 |
| | StableMoE | | 77.7 | 34.3 | 83.9 | 60.8 |
| EC (Topk=2) | | 7.7620 | 81.5 | 39.3 | 88.0 | 75.6 |

*Table 19.* Accuracy performance comparison of USMoE, Token Choice (TC), and Expert Choice (EC) after fine-tuned on various datasets. Higher is better, best results are in **bold**.

| Dataset | PGD | | | | FGSM | | | | SPSA | | | |
|---|---|---|---|---|---|---|---|---|---|---|---|---|
| ViT-MoE (10M) | USMoE | TC | EC | SoftMoE | USMoE | TC | EC | SoftMoE | USMoE | TC | EC | SoftMoE |
| CIFAR-10 | 57.4 | 27.7 | **57.5** | 55.6 | **49.5** | 26.4 | 48.5 | 45.8 | 83.6 | 33.4 | **87.0** | 69.1 |
| CIFAR-100 | **27.5** | 11.9 | 27.0 | 28.6 | **21.5** | 11.1 | 19.3 | 20.6 | **65.8** | 15.8 | 59.1 | 39.3 |
| STL-10 | **41.0** | 37.7 | 39.5 | 39.8 | **36.5** | 35.8 | 34.8 | 35.0 | **64.3** | 44.1 | 61.3 | 49.6 |
| SVHN | 81.0 | 38.4 | 80.5 | **91.3** | **76.0** | 36.1 | 75.4 | 69.2 | **92.5** | 36.3 | 91.8 | 84.4 |
| ImageNet-1K | **24.0** | 1.3 | 22.4 | 13.3 | **12.2** | 1.4 | 11.2 | 9.9 | **13.4** | 1.5 | 12.3 | 10.9 |
| Avg. | **46.2** | 23.4 | 45.4 | 45.7 | **43.3** | 22.2 | 37.8 | 36.1 | **63.9** | 26.2 | 62.3 | 50.7 |

*Table 20.* Robustness evaluation of different ViT-MoE models under adversarial attacks: PGD, FGSM, and SPSA across five datasets. Bold indicates the best performance for each task and attack.

| Transformer-XL(20M) | | Enwik8 | Text8 | WikiText-103 | lm1b |
|---|---|---|---|---|---|
| USMoE (Topk=2) | | **1.18** | **1.20** | **29.20** | **56.90** |
| (Topk=1.5) | | 1.19 | 1.28 | 30.67 | 57.55 |
| TC (Topk=2) | SMoE | 1.20 | 1.29 | 30.16 | 58.00 |
| | SMoE-DR | 1.56 | 1.56 | 58.37 | 93.17 |
| | XMoE | 1.21 | 1.28 | 30.34 | 58.33 |
| | StableMoE | 1.20 | 1.28 | 29.97 | 58.25 |
| EC (Topk=2) | | 1.18 | 1.24 | 29.83 | 58.60 |

*Table 21.* Performance comparison of USMoE, Token Choice (TC), and Expert Choice (EC) across multiple datasets, with BPC on the Enwik8 and Text8 test sets, and perplexity on the WikiText-103 and One Billion Word test sets. Lower values are better, with the best results highlighted in **bold**.

| Dataset | Input length | Batch size | Optimizer | Lr | # Iterations |
|---|---|---|---|---|---|
| enwik8 | 512 | 48 | Adam | 2.5e-4 | 100k |
| text8 | 512 | 48 | Adam | 2.5e-4 | 100k |
| WikiText-103 | 512 | 22 | Adam | 2.5e-4 | 100k |
| One Billion Word | 512 | 11 | Adam | 2.5e-4 | 100k |

*Table 22.* Implementation details for pre-training experiments on enwik8, text8, WikiText-103, and One Billion Word datasets.

| Dataset | Input length | Batch size | Optimizer | Lr | # Epochs |
|---|---|---|---|---|---|
| SST-2 | 512 | 16 | Adam | 1e-4 | 15 |
| SST-5 | 512 | 16 | Adam | 1e-4 | 15 |
| IMDB | 512 | 4 | Adam | 1e-4 | 15 |
| BANKING77 | 512 | 16 | Adam | 1e-4 | 15 |

*Table 23.* Implementation for fine-tuning experiments on downstream tasks.

