# OpenReview forum: "Rethinking Sparse Mixture of Experts from a Unified Perspective"
_ICML.cc/2026/Conference — ICML 2026 regular_

### Official Review · Reviewer_XsRP · 2026-03-06

**Soundness:** 3
**Presentation:** 2
**Significance:** 3
**Originality:** 3
**Overall Recommendation:** 4
**Confidence:** 4

**Summary:**

The paper proposes a unified perspective on Sparse Mixture of Experts (SMoE) by casting routing as a simple linear program with a global budget and introduces USMoE, consisting of (i) a Unified Score that linearly combines row-wise (Token Choice) and column-wise (Expert Choice) normalizations, and (ii) a Unified Mechanism that selects the top-c token–expert pairs globally across the score matrix. The authors claim this design yields more robust routing under noise, mitigates representation collapse, and flexibly supports fractional expert budgets.

**Compliance With Llm Reviewing Policy:**

Affirmed.

**Final Justification:**

The authors provide sufficient and strong responses to address all my concerns. The paper provides limited insights that can inspire readers into future research. The method is relatively simple yet effective, which is a strong technical contribution to address the MoE routing problems, such as token drops and information leakage issues.

I decide to keep my score.

**Key Questions For Authors:**

It will be great if the author provide more details on how the proposed method on auto-regressive task and I also concerning token drop problem if no expert is acitvated.

**Limitations:**

No. The proposed method has potential disadvantages on auto-regressive task such as reasoning and it may also produce token drop problem if a token activated none of experts.

**Strengths And Weaknesses:**

Strength:

1. The framing of token-expert assignment as a single global top-c selection provides a simple, unified view that ties together Token Choice and Expert Choice behaviors. It is interesting that the combination of both view can consistently outperform selected basedline in many task and datasets.
2. The paper provide sufficient experiments and solid clarification on theory to demonstrate the effectiveness of USMOE.
3. nice visualization and clear derivation, but needs a bit of re-organization on overall layout.
4. The idea is easy to grasp

Weakness:

1. Some equations have typos and needs to be revised further (e.g Equation 20, first equation of part 1 of proof Lemma 3.5)
2. The authors critisize Expert-Choice model suffering from auto-regression due to information leakage. The proposed method produce a unified scoring matrix that linearly combine token choice and expert choice matrix. The part of expert choice also breach casuality of autoregression.
 The breach of casuality also reflect on Algorithm 1 Step 4 and 5. The flatten operation also indicate that the routing prior tokens somehow depends on the tokens in the future.
3. Based on point 2, it is not very clear how the proposed operate on reasoning task. It lacks of discrepancy on the inference of proposed method.
4. Many baseline remains large gap to USMOE, it will be great to compare the some latest baseline such as ReMoE (2024)
5. The proposed method stated that replaced Softmax opperator with sigmoid and enforce sparsity on the flatten dimension $ L \times N $. It may lead to token drop problem in practice, which the dropped token activated zero expert. It will be great to have more clarification.

---

> ### Author Rebuttal · Authors · 2026-03-30
>
> Thank you for your constructive comments and insightful suggestions. We would like to address the concern as below:
>
> ```W1: Some equations have typos error (Equation 20) ```
>
> A1: We thank the Reviewer for the suggestion, and will revise typo error in Eqa. 20.
>
> ```W2 & W3: Information leakage issue - Performance on Reasoning```
>
> A2: Figure 7 (Section A.4.4) shows that information leakage in Expert Choice (EC) mainly stems from softmax-based routing, not the selection mechanism. Softmax introduces cross-token dependencies via global normalization, causing interference between tokens. Replacing it with sigmoid removes this coupling by applying independent scoring per entry, significantly reducing leakage. Incorporating this into USMoE improves performance across training-from-scratch, SFT, and zero-shot settings, for both generative and embedding tasks, demonstrating effective mitigation of information leakage.
>
> *Reasoning performance:* On reasoning tasks, USMoE improves accuracy over the original model (TC), increasing the average from 0.706 to 0.721. At the same time, it reduces the average thinking length from 569 to 552 tokens, indicating more efficient reasoning. The results indicate that USMoE mitigates information leakage.
>
> | Model                     | Tasks       | Original (TC) Acc | USMoE Acc | Original (TC) Thinking Tokens | USMoE Thinking Tokens |
> |---------------------------|-------------|-------------------|-----------|-------------------------------|-----------------------|
> | Qwen3-30B-A3B-Thinking    | ARC-C       | 0.584             | 0.600     | 586                           | 565                   |
> |                           | ARC-E       | 0.819             | 0.822     | 353                           | 346                   |
> |                           | BoolQ       | 0.866             | 0.870     | 508                           | 484                   |
> |                           | OBQA        | 0.424             | 0.450     | 630                           | 583                   |
> |                           | PIQA        | 0.812             | 0.842     | 638                           | 664                   |
> |                           | WinoGrande  | 0.729             | 0.739     | 701                           | 671                   |
> |                           | **Average** | **0.706**         | **0.721** | **569**                       | **552**               |
>
>
> ```W4: latest baseline such as ReMoE (2024)```
>
> A3: We report USMoE results under both the original setting (K=8) and with K matched to ReMoE's actual TopK for fair comparison (see table below). USMoE significantly outperforms ReMoE across all tasks, achieving much higher accuracy both at the same TopK=8 setting and under comparable compute (TopK approximate 1.5). When matched to ReMoE's computation (same TopK), USMoE maintains a clear performance advantage, demonstrating better efficiency-accuracy trade-off.
>
>
> | Model | Task | ReMoE | USMoE(K=8) | USMoE(K=ReMoE) | Actual K of ReMoE |
> |-------|------|-------|--------|--------|------|
> | Qwen3-30B-A3B-Instruct | ARC-C | 0.241 | 0.646 | 0.265 | 1.4 |
> |       | ARC-E | 0.258 | 0.846 | 0.352 | 1.3 |
> |       | BoolQ | 0.444 | 0.894 | 0.587 | 1.6 |
> |       | OBQA  | 0.266 | 0.460 | 0.274 | 1.5 |
> |       | PIQA  | 0.515 | 0.810 | 0.574 | 1.3 |
> |       | Wino  | 0.516 | 0.736 | 0.615 | 1.6 |
> |       | Avg   | 0.373 | 0.732 | 0.444 | 1.5 |
>
>
> ```W5-Q1-L1:  token drop problem```
>
> A4: We report the average token dropping rate of Qwen3-30B-A3B-Instruct across six datasets as the table below. EC exhibits non-trivial token dropping (average 3%), which can lead to information loss and degraded performance, whereas both USMoE and TC eliminate token dropping entirely (0% across all tasks), preserving input information while maintaining sparse routing.
>
> | Model | Task | TC | EC | US |
> |------|------|----|----|----|
> | Qwen3-30B-A3B-Instruct(K=8) | ARC-C | 0.00 | .03 | 0.00 |
> |            | ARC-E | 0.00 | 0.03 | 0.00 |
> |            | BoolQ | 0.00 | 0.01 | 0.00 |
> |            | OBQA  | 0.00 | 0.02 | 0.00 |
> |            | PIQA  | 0.00 | 0.04 | 0.00 |
> |            | Wino  | 0.00 | 0.05 | 0.00 |
> |            | Avg   | 0.00 | 0.03 | 0.00 |
>
> *USMoE Token Dropping Theory Results:*
> We theoretically bound the token dropping rate of USMoE by $e^{-K}$, where $K$ is the TopK number of activated experts.
>
> Let $U \in \mathbb{R}^{L \times N}$ be the unified score matrix with $L$ tokens and $N$ experts. USMoE selects the global top-$n$ entries. The marginal probability that an entry is retained is $ \rho = \frac{n}{LN} = \frac{K}{N} $.
>
> A token $\ell$ is dropped if none of its $N$ entries are retained. Under an independence approximation: $ \Pr(\mathcal{D}_\ell) = (1-\rho)^N = \left(1-\frac{K}{N}\right)^N $.
>
> Using the inequality $ (1-x)^N \le e^{-Nx} $ for $x \in [0,1]$, we obtain $ \Pr(\mathcal{D}_\ell) \le e^{-K} $.
>
> In practice, $K=8$, so $ \Pr(\mathcal{D}_\ell) \le e^{-8} \approx 0.0003 $.

---

> > ### Author Rebuttal · Reviewer_XsRP · 2026-04-01
> >
> > Thank you for your response. I will maintain my original score.

---

### Official Review · Reviewer_zYRB · 2026-03-08

**Soundness:** 3
**Presentation:** 3
**Significance:** 3
**Originality:** 2
**Overall Recommendation:** 4
**Confidence:** 3

**Summary:**

The paper suggests that existing SMoE (Sparse Mixture of Experts) fall into two categories: Token Choice (TC), which routes each token to a fixed number of experts, and Expert Choice (EC), which assigns a fixed number of tokens to each expert.
Authors argue that existing TC and EC has its limitation, and both methods are constrained special cases of a global optimization: maximize total token-expert compatibility scores under a budget constraint.
Authours propose USMoE which introduces two components, where one is unified score that linearly combines row-wise and column-wise score, and the other is unified mechanism that performs global TopK selection from compatibility score.
Theoretical results show robustness advantages over TC under noisy tokens and reduced representation collapse.
Extensive experiments on diverse data settings, multiple domains show that it is effective.

**Compliance With Llm Reviewing Policy:**

Affirmed.

**Final Justification:**

after rebuttal, i decided to keep my score.

**Key Questions For Authors:**

1.
I think global TopK over the compatibility T×N score matrix still remove all per-token and per-expert load constraints.
This might still lead to expert load imbalance, where some experts may receive disproportionately many tokens while others are starved.
Thus, could you provide how many tokens does each expert receive on average, and what is the variance across experts?

2.
Could you provide more details or insight on using sigmoid on EC (such as how it performs better than softmax)?
The experiment on Figure 7 shows promising results, but I am still confused about the insight behind this.
Although it is on the Appendix, I think the contribution of the paper would be more strenghtened with proper analysis.

3.
Could you provide ablations on each unified score and unified mecahnism score?
For example, the setting only with unified score, only with unified mechanism, and both would help readers to understand the contriution of each component clearly.

**Limitations:**

1.
Load balancing under global TopK is not analyzed.
Without per-token or per-expert constraints, expert starvation and utilization imbalance could degrade inference efficiency on parallel hardware.

2.
The connection between LP optimality and downstream model performance is not formally established.

3.
Ablation between the unified score and unified mechanism is missing, making it unclear which component drives the observed gains.

**Strengths And Weaknesses:**

## Strengths

-
The motivation is clear where existing TC and EC has its own limitations (e.g., noisy token, representation collapse, token drop + information leak by using sigmoid instead of softmax), and authors suggest practical method to handle such limitations.

-
The LP formulation (Eq. 2) that unifies Token Choice and Expert Choice as constrained special cases of the same global optimization problem is clean and insightful.
Proposition 3.3 correctly establishes that global TopK is the optimal solution under a total budget constraint, providing a principled foundation for the routing strategy.

-
Extensive experiments on diver setting, datasets, settings support the method.

-
The robustness analysis is convincing.
Lemma 3.5 provides theoretical grounding for why global competition filters noisy tokens, and Figure 3 empirically confirms this.


## Weaknesses

-
The LP formulation is clean, but the connection between optimizing the LP objective (total compatibility score) and improving model performance is not formally established.
Even Lemma 3.5 only bounds misrouting probability under noise, where the theoretical justification for performance gains remains indirect.

-
Global TopK removes all per-token and per-expert load constraints, which may still lead to expert load imbalance.
The paper provides no analysis of expert utilization distribution (e.g., average tokens per expert, variance), making it unclear whether the routing is practically deployable on parallel hardware.

-
The contribution of the unified score and unified mechanism are not ablated separately.
Since the unified score alone already integrates TC and EC perspectives, it is unclear how much of the gain comes from each component independently.

---

> ### Author Rebuttal · Authors · 2026-03-30
>
> Thank you for your time and your constructive feedback. We would like to address the concerns as follows:
>
> ```W1 & L2: connection between optimizing the LP objective (total compatibility score) and improving model. Lemma 3.5 only bounds misrouting.```
>
> A1: The LP formulation (Prop. 3.3) shows that TC (SMoE) is suboptimal, while USMoE achieves the optimal solution under a global budget constraint. Moreover, Lemma 3.5 proves that USMoE has lower or equal misrouting probability than TC, supporting its stronger performance under corrupted settings (Section 4).
>
> ```W2 & Q1 & L1: expert load imbalance issue. tokens does each expert receive on average```
>
> A2: As TC[1], we incorporate a load-balancing loss to address expert imbalance.
>
> | Model                 | Data    | Metric           | TC   | USMoE |
> |----------------------|---------|------------------|------|-------|
> | Transformer-XL (20M) | Enwik8  | Load-Balancing Loss   | 1.64 | 0.98  |
> |                      |         | BPC              | 1.20 | 1.18  |
>
> On Transformer-XL (20M) with Enwik8, USMoE reduces the balancing loss (0.98 vs. 1.64) as above table, indicating more uniform token distribution across experts. This suggests improved load balancing with lower variance in expert utilization. Moreover, USMoE achieves an improvement in BPC (1.18 vs. 1.20), demonstrating that better balancing can enhance language modeling performance.
>
> ```W3 & Q3 & L3: unified score and unified mechanism are not ablated separately```
>
> A3: We thank the reviewer for this suggestion. Below table provides an ablation that disentangles the contributions of the Unified Mechanism (UM) and Unified Score (US). The results show that both components contribute consistently across tasks, with UM accounting for the majority of gains on average (60%), while the US provides complementary improvements (40%). This demonstrates that the two components are both necessary and mutually reinforcing.
>
> | Model                              | Task       | USMoE | Original (TC) | USMoE (UM only) | UM contrib | US contrib |
> |-----------------------------------|-----------|-------|---------------|----------------------------|------------|------------------------|
> | Qwen3-30B-A3B-Instruct (0-shot)   | ARC-C     | 0.646 | 0.631         | 0.645                      | 0.96       | 0.04                   |
> |                                   | ARC-E     | 0.846 | 0.838         | 0.844                      | 0.74       | 0.26                   |
> |                                   | BoolQ     | 0.894 | 0.886         | 0.893                      | 0.87       | 0.13                   |
> |                                   | OBQA      | 0.460 | 0.454         | 0.458                      | 0.58       | 0.42                   |
> |                                   | PIQA      | 0.810 | 0.805         | 0.807                      | 0.34       | 0.66                   |
> |                                   | WinoGrande| 0.736 | 0.733         | 0.733                      | 0.11       | 0.89                   |
> |                                   | Average   | 0.732 | 0.724         | 0.730                      | 0.60       | 0.40                   |
>
> ```Q2: provide more details or insight on using sigmoid on EC```
>
> A4: The Expert Choice (EC) formulation with softmax suffers from information leakage and token dropping, which can degrade language modeling performance. As shown in Fig. 7, the sigmoid-based EC achieves lower BPC and significantly reduces token dropping. This suggests that sigmoid mapping mitigates these issues and is more suitable for scaling EC in large language models.
>
> **Reference**
>
> [1] OLMoE: Open Mixture-of-Experts Language Models. ICLR. 2025

---

> > ### Author Rebuttal · Reviewer_zYRB · 2026-04-03
> >
> > Thank the authors for the time and effort during the rebuttal phase, and I think most of the concerns are resolved well.
> >
> > However, I still have a question on response to [W1 & L2].
> > I understand from the paper that the LP formulation is suboptimal when using TC, while USMoE achieves the optimal solution under the budget including lower or equal misrouting probability than TC.
> > But my original point was about the insight behind the connection between optimizing the LP objective (total compatibility score) and improving model performance.
> >
> > That being said, why is **optimizing LP objective (increasing compatibility score) beneficial in the first place** (not that USMoE is the optimal solution to the LP objective)?

---

> > > ### Author Response · Authors · 2026-04-03
> > >
> > > Dear Reviewer zYRB,
> > >
> > > Thank you for your thoughtful feedback and for acknowledging our previous responses. We also appreciate your follow-up questions regarding the connection between optimizing the LP objective and model performance, and we address them below:
> > >
> > > ```FQ1:  insight behind the connection between optimizing the LP objective (total compatibility score) and improving model performance.```
> > >
> > > A1: The LP objective is not an abstract quantity but follows the core assumption of Sparse Mixture of Experts(SMoE): tokens are routed to Top-K experts with highest token-expert compatibility [1]. The compatibility score (after normalization) measures token-expert affinity, so maximizing this objective encourages selecting more suitable experts, improving expert specialization and model performance. Our method (USMoE) builds directly on this principle and extends it to a globally optimal 2D selection.
> > >
> > > **Reference**
> > >
> > > [1] Outrageously Large Neural Networks: The Sparsely-Gated Mixture-of-Experts Layer. ICLR. 2017

---

### Official Review · Reviewer_3fHR · 2026-03-09

**Soundness:** 2
**Presentation:** 2
**Significance:** 3
**Originality:** 3
**Overall Recommendation:** 4
**Confidence:** 2

**Summary:**

The paper introduces a unified framework, USMoE, which models expert assignment as a global optimal matching problem, bridging Token Choice (TC) and Expert Choice (EC) in Sparse Mixture of Experts (SMoE). It computes a "Unified Score" (a weighted sum of TC and EC routing scores) and employs a "Unified Mechanism" (a global or sequence-level Top-K selection) to improve routing robustness and mitigate representation collapse. The authors demonstrate the empirical effectiveness of USMoE across large language models and vision tasks in both training-free and training-from-scratch scenarios.

**Compliance With Llm Reviewing Policy:**

Affirmed.

**Final Justification:**

In summary, while I appreciate the authors' impressive experimental results, I still find certain aspects of the paper unclear. Despite the rebuttal, I remain unconvinced regarding the theoretical correctness and robustness of the proposed method. Therefore, I will maintain my original score.

**Key Questions For Authors:**

1. Since Proposition 3.3's optimality proof relies on the raw dot-product similarity matrix $S$, how does this proof remain valid when the global Top-K is applied to the non-linearly transformed Unified Score matrix $U$ (as shown in Algorithm 1)?
2. Regarding Appendix A.1.2, could you provide a more rigorous mathematical explanation for why having more terms ($2n \gg n$) definitively justifies an increase in the Jacobian matrix's rank and proves the mitigation of representation collapse?
3. Given that MoEE is designed for text embeddings, what is the rationale for using it as a baseline for reasoning tasks in Table 1? Have you considered comparisons with more standard advanced routing approaches (e.g., SoftMoE for text, or models with shared experts)?
4. Could you clarify the exact perturbation methods used for the "corrupted" settings? Specifically, how is continuous Gaussian noise applied to discrete tokens in the zero-shot setting (Section 4.1.1), and how does this compare to the "AAA" replacement in the SFT setting (Appendix A.4.1)? Additionally, given USMoE's claimed robustness to noise, how does it perform on context-heavy evaluations like Needle-In-A-Haystack?
5. Does the global assignment strategy in USMoE lead to expert load imbalance during training-from-scratch (especially large scale language models)? Was any auxiliary load-balancing loss employed? Furthermore, how does the global sorting operation affect training/inference throughput (communication overhead) when scaling to large batch sizes and sequence lengths?

**Limitations:**

The authors have not adequately discussed the limitations of their method in the main text. Specifically, as a global assignment strategy, USMoE inevitably faces a high risk of load imbalance (with some experts overloaded and others underutilized). Additionally, the global sorting operation (Flatten + Top-K) could introduce significant communication and computational overhead at very large batch sizes. I recommend that the authors include a candid discussion of these practical deployment constraints, particularly regarding load balancing and scalability overhead.

**Strengths And Weaknesses:**

**Strengths:**
1. Originality: The perspective of shifting from local Token/Expert Choice to a global or sequence-level joint assignment is highly intuitive and offers an interesting unified view of MoE routing.
2. Significance: The experimental evaluation is extensive, covering both text and vision modalities, training-free (plug-and-play) settings, and training-from-scratch scenarios.

**Weaknesses:**
1. Confusing writing: The writing and definitions can be somewhat confusing. Specifically, the transition from Equation 6 (raw similarity) to Definition 3.1 (row and column softmax combination score) is confusing and lacks clear justification. Moreover, the boundary between "Unified Score" and "Unified Mechanism" seems artificial.
2. Theory-Algorithm gap: Proposition 3.3 proves that global Top-K maximizes the total similarity on the raw dot-product matrix. However, Definition 3.1 and Algorithm 1 apply this global Top-K to the "Unified Score," which is a linear combination of row-wise and column-wise softmax scores. Operating on this matrix breaks the linear programming optimality guarantee presented in the text, creating a disconnection between the theoretical claims and the actual algorithm.
3. Mathematical justification: The theoretical justification regarding the Jacobian matrix and representation collapse in Appendix A.1.2 (Proof of Proposition 3.3) is technically unconvincing. The authors attempt to prove that USMoE mitigates collapse by comparing the number of terms in the expansion ($2n \gg n$). However, the rank of a Jacobian matrix—which determines the dimension of the representation space—depends on the linear independence of the vectors, not simply the arithmetic count of terms.
4. Questionable baseline choice: In Table 1, MoEE is used as a core baseline. However, MoEE is specifically designed for extracting sentence embeddings, not as a routing strategy for autoregressive text generation, making this comparison confusing and uninformative.
5. Inconsistent experimental details: The term noisy tokens and corrupted setting is confusing. The description of the robustness experiments contains conflicting details. Section 4.1.1 claims the injection of "15% random Gaussian noise," whereas Appendix A.4.1 states that words are randomly replaced with a generic token "AAA".
6. Unaddressed load balancing issue: A global Top-K routing strategy inherently risks severe load imbalance (e.g., some experts being overloaded while others are starved). The paper does not adequately discuss how this is resolved during the training-from-scratch experiments (e.g., whether an auxiliary loss is used). This omission raises concerns about the practical scalability of the method to billion-parameter models.

---

> ### Author Rebuttal · Authors · 2026-03-30
>
> We thank the reviewer for the valuable feedback and would like to address concerns as below:
>
> ```W1: The writing and definitions confusing```
>
> A1: We apologize for the ambiguity in the definition of $S$ in Def. 3.1. $S$ denote for compatibility score after distribution mapping (eg., softmax). To avoid notation conflicts, we will revise Def. 3.1.
>
> ```W2: Theory-Algorithm gap: matrix breaks the linear programming```
>
> A2: There are two misunderstandings as below:
>
> *   *Compatibility Score $S$ (Prop. 3.3)*: $S$ is the output of a non-linear transformation (eg:softmax) applied to router logits. For the notation conflicts, as above, we will revise notations for Def 3.1.
> *   *Linear Programing formulation (Eq. 2)*: THe objective is linear with respect to $x$ (expert selection mask) not $S$. Thus, the linearity is preserved regardless of the non-linear transformation of $S$.
>
> *Q1: Prop. 3.3 with non-linearly transformed Unified Score matrix?*
>
> A: Score $S$ is output of non-linear transformation in Prop. 3.3. If $S$ is dot product similarity, Prop. 3.3 still holds. Key result depends on ranking induced by score matrix $S$, and common non-linear transformation such as Softmax are monotonic ($Rank(R) = Rank(Softmax(R)$). Thus, Prop. 3.3 holds for both non-linear transformation (eg: softmax) and dot product similarity.
>
> ```W3 and Q2: representation collapse in Appendix A.1.2 (Proof of Proposition 3.3) is technically unconvincing ```
>
> A3: [1] raises the representation collapse by examining the Jacobian matrix. TC (SMoE) Jacobian can be expressed linear combination of $N$ expert embedding, projecting the input $x \in R^d$ into $N$ dimension space. When $N << d$, this leads to the representation collapse.
>
> In contrast, USMoE combines TC and EC scores, yielding Jacobian that spans a linear combination of $2N$ experts embeddings. In practice, $2N << d$ does not always hold (eg: N=128, d=1024), resulting in a richer representation space. Thus, USMoE more effectively to address the representation collapse compared to TC.
>
>
> ```W4 and Q3: Questionable baseline choice: In Table 1```
>
> A4: Our core baselines are TC (SMoE) and EC. We also include MoEE for two reasons: (1) it is a training-free method, ensuring a fair comparison with USMoE; and (2) it demonstrates that router signals encode semantic information, which aligns with our finding that EC router scores provide strong semantic representations.
>
> Additionally, following Reviewer XsRP's suggestion, we include ReMoE [2] as an additional baseline. Detailed results are provided in the corresponding section addressing Reviewer XsRP.
>
> ```W5 and Q4: Inconsistent experimental details```
>
> A5: In Section 4.1.1 (training-free), we inject 15% random Gaussian noise to evaluate robustness. Additionally, we conduct experiments on supervised fine-tuning (SFT) using the Alpaca dataset (Section A.4.1), where we introduce noise tokens ("AAA"). These additional experiments demonstrate the effectiveness of our method in SFT settings, while Section 4.1.1 focuses on its training-free capability.
>
> *Corrupt*: For the training-free (zero-shot) setting (Section 4.1.1), we inject Gaussian noise into the outputs of the Multi-Head Attention (MHA), which serve as inputs to the MoE layers. For the SFT experiments (Section A.4.1), we introduce noise at the text level by randomly replacing words with the token "AAA".
>
> *NIAH*: USMoE outperforms both TC and EC in the Needle-in-a-Haystack evaluation, achieving perfect retrieval accuracy (100%) compared to 50% for TC and 10% for EC. This illustrates the superior long-context retrieval capability of USMoE under challenging, distraction-heavy settings.
>
>
> | Model                         | Context Length | Doc Depth | Method         | Acc of Retrieval |
> |------------------------------|---------------|-----------|----------------|------------------|
> | Qwen3-30B-A3B-Instruct       | 16.5K         | 50%       | Original (TC)  | 50%              |
> |                              |               |           | EC             | 10%              |
> |                              |               |           | USMoE          | **100%**             |
>
> ```W6 and Q5: Unaddressed load balancing issue```
>
> A6:
> *   Training-free setting: No Load-balancing Loss
> *   Training setting: Following TC[3], we incorporate a load-balancing loss to address expert imbalance.
>
> *Communication Overhead*:
> As discussed in our complexity analysis (Response to Reviewer XAeK), USMoE operates per sample and does not introduce additional cross-device communication beyond the standard expert dispatch in conventional SMoE. While USMoE involves larger TopK reductions, this only affects constant factors and has negligible impact on throughput in practice.
>
> **Reference**
>
> [1] On the Representation Collapse of Sparse Mixture of Experts. NeurIPS. 2022
>
> [2] ReMoE: Fully Differentiable Mixture-of-Experts with ReLU Routing. ICLR. 2025
>
> [3] OLMoE: Open Mixture-of-Experts Language Models. ICLR. 2025

---

> > ### Author Rebuttal · Reviewer_3fHR · 2026-04-02
> >
> > I acknowledge that I have read the author rebuttal. The authors have clarified experimental inconsistencies (noise injection settings) and confirmed load-balancing mechanisms during training, which addresses practical implementation concerns. However, core theoretical issues remain insufficiently addressed. Specifically, the justification for Proposition 3.3's optimality when applied to the non-linear Unified Score matrix relies on a monotonicity argument that does not fully bridge the gap between maximizing the unified score and optimizing semantic similarity. Additionally, the mathematical proof regarding Jacobian rank (Appendix A.1.2) remains heuristic rather than rigorous. Since these theoretical claims are central to the paper's contribution, I require further clarification on how these gaps impact the validity of the proposed framework before adjusting my score.

---

> > > ### Author Response · Authors · 2026-04-02
> > >
> > > Dear Reviewer 3fHR,
> > >
> > > Thank you for your thoughtful feedback and for acknowledging our previous responses. We also appreciate your follow-up questions regarding Proposition 3.3 and Lemma 3.7, and we address them as below:
> > >
> > > ```FQ1: optimality when applied to the non-linear Unified Score matrix relies on a monotonicity argument that does not fully bridge the gap between maximizing the unified score and optimizing semantic similarity ```
> > >
> > > A1: We emphasize that USMoE does not aim to optimize raw semantic similarity; instead, it directly optimizes a unified score that incorporates normalization and balancing effects. Proposition 3.3 holds for this score by construction, ensuring consistency between the objective and the algorithm. Furthermore, this score mitigates token-level dominance and expert imbalance, leading to improved empirical performance, as also reflected in reduced token dropping and more stable expert utilization.
> > >
> > > ```FQ2: Additionally, the mathematical proof regarding Jacobian rank (Appendix A.1.2) remains heuristic rather than rigorous.```
> > >
> > > A2: We follow prior work XMoE(Chi et al., NeurIPS 2022) to analyze representation collapse via the Jacobian. In SMoE, the Jacobian can be expressed as a linear combination of expert embeddings, which may lead to a low-rank structure when the number of experts is small relative to the representation dimension.
> > >
> > > In contrast, USMoE combines token-wise and expert-wise routing signals, yielding a Jacobian composed of two distinct components. While we agree that rank depends on linear independence rather than the number of terms, the key point is that these two components are induced by different normalization mechanisms, producing non-collinear coefficient vectors in practice. Under mild assumptions that $S_e$ and $S_t$ are not perfectly correlated, the resulting Jacobian spans a strictly larger subspace in expectation.
> > >
> > > Therefore, USMoE mitigates representation collapse by increasing the effective rank of the Jacobian, and we will clarify this assumption and its implications in the revision.

---

### Official Review · Reviewer_XAeK · 2026-03-12

**Soundness:** 3
**Presentation:** 3
**Significance:** 3
**Originality:** 3
**Overall Recommendation:** 4
**Confidence:** 4

**Summary:**

The paper unifies Token Choice and Expert Choice under a constrained routing formulation and proposes USMoE for joint token-expert selection. It introduces a unified score and a unified routing mechanism, and reports results across language and vision settings, including zero-shot, training-from-scratch, and plug-in evaluations. The paper also discusses routing under corrupted inputs and flexible Top-K selection.

**Compliance With Llm Reviewing Policy:**

Affirmed.

**Final Justification:**

The author resolved the concern after rebuttal. I maintain a positive review of this paper.

**Key Questions For Authors:**

See weaknesses.

**Limitations:**

Yes.

**Strengths And Weaknesses:**

Strengths:

1.	Well-motivated problem formulation.
The paper identifies a clear limitation of existing MoE routing, namely that Token Choice and Expert Choice make routing decisions along only one dimension. The proposed method is well aligned with this problem.
2.	Conceptually more principled than a simple heuristic combination.
Rather than naively combining TC and EC, the paper provides a unified view of both and proposes a joint routing mechanism based on that perspective.
3.	Consistent empirical improvements over prior routing methods.
The reported results show that the proposed method generally outperforms TC/EC and related baselines across multiple settings.






Weaknesses:

1.	Limited systems-efficiency analysis. The proposed routing mechanism relies on global selection over the token-expert score matrix, which may introduce additional routing overhead compared with standard TC/EC. However, the paper does not provide sufficient analysis of deployment cost.

2.	The “optimality” is limited to the paper’s formulation. While the unified routing is theoretically motivated, the claimed optimality mainly holds under the proposed constrained formulation. It is less clear whether this translates to globally optimal routing behavior in practical large-scale MoE settings.

3.	The method improves routing selection, but may not fully address the root cause. The paper mainly improves how token-expert pairs are selected given a score matrix. It is less clear whether the method fundamentally addresses the quality of the routing scores themselves or the deeper cause of routing degeneration.

4.	More discussion on scalability would be helpful. Since the method is evaluated on several settings, it would strengthen the paper to include a clearer discussion of how the routing mechanism scales with larger numbers of tokens and experts.

---

> ### Author Rebuttal · Authors · 2026-03-30
>
> We sincerely thank the reviewer for the constructive feedback and would like to address your concerns as follows:
>
> ```W1: Limited systems-efficiency analysis.```
>
> A1: USMoE and Token Choice (TC or SMoE) share the same asymptotic complexity.
>
> Router complexity is the sum of the score computation and selection cost.
>
> *TC*: $O(BLDN + BLN)+ O(B\cdot L \cdot \mathrm{TopK}(N, k))$.
>
> *USMoE*: $O(BLDN + BLN) + O(B \cdot \mathrm{TopK}(L \cdot N, L \cdot k))$
>
> ($B, L, D, N$ are batch size, sequence length, model dimension, and number of experts, respectively)
>
> Under standard near-linear selection part implementations, both are dominated by the score computation complexity $O(BLDN)$.
>
>
> ```W2: The “optimality” is limited to the paper's formulation.```
>
>
> A2: We clarify that "optimality" refers to the optimal expert selection solution for the optimization problem defined in Eq. 2. Our mechanism is agnostic to the choice of router (e.g., Softmax, Sigmoid, or ReLU), and thus we claim optimal selection under a fixed set of router scores.
>
> ```W3: The method improves routing selection, but may not fully address the root cause. ```
>
> A3: Our Unified Score mitigates representation collapse [1], while our Unified Mechanism addresses noise sensitivity (Lemma 3.5, Fig. 3) and suboptimal selection (Prop. 3.3). Thus, USMoE outperforms TC in both training-free and training settings.
>
> ```W4: More discussion on scalability ```
>
> A4: Beyond comparing USMoE with TC across model sizes (20M - 30B parameters), we further evaluate scalability under constrained computation (TopK = 1,2,4,6,8). As shown in the table, USMoE consistently outperforms TC across all TopK settings, demonstrating its clear advantage.
>
> | Setting | Model                     | Data  | TopK | TC (Original) | USMoE |
> |---------|---------------------------|-------|------|---------------|-------|
> | Corrupt | Qwen3-30B-A3B-Instruct    | BoolQ | 1    | 0.418         | **0.455** |
> |         |                           |       | 2    | 0.483         | **0.493** |
> |         |                           |       | 4    | 0.489         | **0.693** |
> |         |                           |       | 6    | 0.490         | **0.784** |
> |         |                           |       | 8    | 0.600         | **0.808** |
>
>
> **References**
>
> [1] On the Representation Collapse of Sparse Mixture of Experts. NeurIPS. 2022

---

> > ### Author Rebuttal · Reviewer_XAeK · 2026-04-01
> >
> > Thank you for your response. After careful consideration, I will maintain my original score.

---

### Decision · Program_Chairs · 2026-04-30

**Decision:**

Accept (regular)

**Comment:**

This paper proposes USMOE, a unified framework for Sparse Mixture of Experts (SMoE) routing. It re-frames Token Choice (TC) and Expert Choice (EC) as constrained special cases of a global linear programming (LP) problem. The core technical contribution is a two-part mechanism: a "Unified Score" that combines row-wise and column-wise normalized affinities, and a "Unified Mechanism" that selects the globally highest-scoring token-expert pairs based on a total budget. The paper provides theoretical arguments for its robustness against noise and representation collapse, supported by extensive experiments.

This paper addresses a clear and well-motivated problem in the SMoE literature: the inherent limitations of strictly enforcing per-token or per-expert budgets. The proposed LP perspective is a conceptually elegant and principled way to unify existing methods.

All the reviewers give the recommendation of Weak Accept and acknowledged that the authors' rebuttal addressed most of the concerns. Some concerns still remain, e.g., revision of theoretical claims, ablation studies, and explicit discussion of limitations. I recommend the authors solve these issues in the final version.

Given that the reviewers reached agreement on the acceptance of this paper, I also vote for acceptance.